
# Source apportionment of fine particulate matter in Houston, Texas: Insights to secondary organic aerosols

Ibrahim M. Al-Naiema[1], Anusha P. S. Hettiyadura[1], Henry W. Wallace[2], Nancy P. Sanchez[2], Carter J. Madler[1], Basak Karakurt Cevik[2,3], Alexander A.T. Bui[2], Josh Kettler[1], Robert J. Griffin[2], and Elizabeth A. Stone[1,4]

[1] Department of Chemistry, University of Iowa, Iowa City, IA, 52242, USA
[2] Department of Civil and Environmental Engineering, Rice University, Houston, TX, 77005 USA
[3] Department of Energy Systems Engineering, Faculty of Engineering, Yalova University, Yalova, 77100, Turkey
[4] Department of Chemical and Biochemical Engineering, University of Iowa, Iowa City, IA, 52242, USA

*Correspondence to*: Elizabeth A. Stone (betsy-stone@uiowa.edu)

**Abstract.**

Online and offline measurements of fine particulate matter (PM) near the urban and industrial Houston Ship Channel in Houston, Texas, USA during May 2015 were utilized to characterize its chemical composition and to evaluate the relative contributions of primary, secondary, biogenic, and anthropogenic sources. Aerosol mass spectrometry (AMS) on non-refractory $PM_1$ (PM ≤ 1 µm) indicated major contributions from sulfate (averaging 50%), organic aerosol (OA, 40%), and ammonium (14%). Positive matrix factorization (PMF) of AMS data categorized OA on average as 22% hydrocarbon-like organic aerosol (HOA), 29% cooking influenced semi-volatile oxygenated organic aerosol (CI-SV-OOA), and 48% low-volatility oxygenated organic aerosol (LV-OOA), with the latter two sources indicative of secondary organic aerosol (SOA). Chemical analysis of $PM_{2.5}$ (PM ≤ 2.5 µm) filter samples agreed that organic matter (35%) and sulfate (21%) were the most abundant components. Organic speciation of $PM_{2.5}$ organic carbon (OC) focused on molecular markers of primary sources and SOA tracers derived from biogenic and anthropogenic volatile organic compounds (VOC). The sources of $PM_{2.5}$ OC were estimated using molecular marker-based positive matric factorization (MM-PMF) and chemical mass balance (CMB) models. MM-PMF resolved 9 factors that were identified as diesel engines (11.5%), gasoline engines (24.3%), non-tailpipe vehicle emissions (11.1%), ship emissions (2.2%), cooking (1.0%), biomass burning (BB, 10.6%), isoprene SOA (11.0%), high-$NO_x$ anthropogenic SOA (6.6%), and low-$NO_x$ anthropogenic SOA (21.7%). Using available source profiles, CMB apportioned 41% of OC to primary fossil sources (gasoline engines, diesel engines, and ship emissions), 5% to BB, 15% to SOA (including 7.4% biogenic and 7.6% anthropogenic), and 39% to other sources that were not included in the model and are expected to be secondary.

This study presents the first application of *in situ* AMS-PMF, MM-PMF, and CMB for OC source apportionment and the integration of these methods to evaluate the relative roles of biogenic, anthopogenic, and BB-SOA. The three source apportionment models agreed that ~50% of OC is associated with primary emissions from fossil fuel use, particularly motor vehicles. Differences among the models reflect their ability to resolve sources based upon the input chemical measurements, with molecular marker-based methods providing greater source specificity and resolution for minor sources. By combining results from MM-PMF and CMB, BB was estimated to contribute 11% of OC, with 5% of primary emissions and 6% BB-SOA. SOA was dominantly anthropogenic (28%) compared to biogenic (11%) or BB-derived. The three-model approach demonstrates significant contributions of anthropogenic SOA to fine PM. More broadly, the findings and methodologies presented herein can be used to advance local and regional understanding of anthropogenic contributions to SOA.



# 1 Introduction

Organic aerosol (OA) comprises a significant fraction of atmospheric particulate matter (PM) in urban environments (Aiken et al., 2009; Cao et al., 2004; Fraser et al., 2002). Secondary organic aerosol (SOA), formed in the atmosphere through the chemical transformation of volatile organic compounds (VOCs), is a major source of organic aerosol mass (Kroll and Seinfeld, 2008; Henze et al., 2008). Current knowledge of the precursors, mechanisms of formation, and properties of SOA is incomplete, leaving major gaps in understanding of exactly how, and to what extent, SOA affects air quality and climate (Foley et al., 2010). In particular, the roles of natural and anthropogenic precursors to SOA are highly uncertain and variable: results of some published studies indicate dominance (>90%) of biogenic precursors like isoprene (Hallquist et al., 2009), while others studies highlight the importance of anthropogenic VOC (>30%), such as benzene and toluene (Volkamer et al., 2006; Henze et al., 2008). In this study, a measurement-based approach is taken to evaluate the relative contributions of biogenic and anthropogenic VOC to SOA, and their role in relation to primary PM sources, in an urban location in Houston, TX.

Source estimation of SOA in the atmosphere is challenging due to the complexity of precursors to and chemical reactions that form it (Hallquist et al., 2009). Model predictions of SOA rely on knowledge of VOC abundance, product volatility, and SOA yields from chamber studies (Seinfeld and Pankow, 2003; Chan et al., 2009; Donahue et al., 2006). Predictions undergo continuous improvement as knowledge of SOA precursors and formation pathways evolves (Robinson et al., 2007; Ng et al., 2007). Measurement-based approaches can be used to provide ground-truthing for model predictions.

The SOA tracer method estimates SOA contributions to ambient organic carbon (OC) or organic aerosol (OA) through measurements of SOA tracers for VOC (i.e., isoprene, α-pinene, toluene, or naphthalene) using the tracer-to-SOA mass fraction obtained from laboratory chamber experiments (Kleindienst et al., 2007; Kleindienst et al., 2012). This approach is useful in identifying and estimating SOA contributions from SOA precursors at receptor sites and can be used in combination with other organic molecular markers in the source apportionment of OC (Lewandowski et al., 2008). The SOA-tracer method, however, is limited to a handful of VOC precursors and should be further expanded to represent the broader diversity of VOC precursors to SOA.

Receptor models are widely applied for the source apportionment of ambient PM (Belis et al., 2013) and provide valuable information to support air quality management (Hopke, 2016). Among these models, molecular marker-based chemical mass balance (CMB) modeling apportions PM or OC measured at the receptor based on least-squares solution to the linear combination of source profiles and their relative contributions to fit ambient measurements (Watson et al., 1984). Accurate solutions for CMB apportionment largely depend on the representativeness of the profiles to the receptor site. CMB has been successful in apportioning the carbonaceous PM to primary sources for which profiles are available (Lough et al., 2007; Schauer et al., 2002; Simoneit et al., 1999; Rogge et al., 1998). However, the CMB model is often unable to apportion OC for which sources are unknown or not well defined (Stone et al., 2009; Sheesley et al., 2017). Even with the incorporation of SOA tracers into CMB modeling (following the previously described SOA-tracer method), a significant fraction of OC remains unapportioned, suggesting that better representation of SOA is needed in this model (Stone et al., 2009). Molecular marker-based positive matrix factorization (MM-PMF) does not require source profiles and instead decomposes ambient measurements into factors and factor contributions that need to be interpreted in order to identify the source types, based on the knowledge of source signatures (EPA-PMF, 2014). MM-PMF requires a large sample size (60-200) to provide a statistically meaningful solution (Jaeckels et al., 2007). Studies that have compared CMB and MM-PMF



results generally show agreement in their estimation for the primary sources, but systematically give higher MM-PMF source estimates than CMB (Shrivastava et al., 2007; Jaeckels et al., 2007).

In recent years, aerosol mass spectrometry (AMS) has been widely used to characterize the OA of non-refractory submicron PM (NR-PM$_1$) and is related to source types using AMS-PMF (Paatero and Tapper, 1994; Ulbrich et al., 2009).
This approach overcomes the complexity and challenges associated with the quantification of the organic species in atmospheric matrixes (Seinfeld and Pankow, 2003; Goldstein and Galbally, 2007) and apportions OA into factors based on their mass fragmentation fingerprints (Zhang et al., 2011). The high time-resolution of AMS enables the identification of diurnal source variations. However, ambiguity can arise in apportioning sources with similar mass fragmentation fingerprints such as cooking and vehicular emissions (Mohr et al., 2009) and with specifying contributing sources.
Houston, TX, an industrial costal city, experiences elevated traffic-related air pollution and VOC emissions from petroleum facilities (Zhang et al., 2017; Buzcu and Fraser, 2006). The urban and industrial areas near the Houston Ship Channel (HSC) have been the subject of source apportionment of VOCs (Xie and Berkowitz, 2006; Buzcu and Fraser, 2006; Dechapanya et al., 2004) and PM. Sullivan et al. (2013) apportioned PM$_{2.5}$ using metals data from 2005 to 2012 with PMF near the HSC and reported major contributions from secondary inorganic sources (33.9% ammonium sulfate and 4.2%
ammonium nitrate), followed by vehicles (17.5% light duty and 4.8% heavy duty) and crustal elements (11.9% calcium sulfate and 6.3% crustal elements), with minor contributions from fires, sea salt and oil combustion. Using molecular marker-based CMB, Fraser et al. (2003) apportioned PM$_{2.5}$ to vehicle emissions (30% from gasoline and diesel), road dust (11%), fuel oil combustion (7%), meat cooking (6%), wood combustion (2%), and vegetative detritus (2%). Also using molecular-marker based CMB, Buzcu et al. (2006) apportioned 49% of PM$_{2.5}$ OC near HSC, predominantly to vehicle
emissions (36% of OC). Applications of AMS-PMF near the HSC suggested that secondary sources contribute 55-68% of OA, but did not distinguish between biogenic and anthropogenic precursors (Cleveland et al., 2012; Wallace et al., 2018) . These source apportionment studies indicate significant influences on PM from motor vehicles and secondary reactions, but lack understanding on the precursors to SOA. Combining these apportionment techniques would be helpful to gain better insights about the composition of organic aerosol (OA) and provide more accurate source characterization.
In this work, we report a compositional analysis and source characterization of the PM$_{2.5}$ and non-refractory PM$_1$ near the HSC in May 2015. An Aerodyne high-resolution time-of-flight AMS (HR-ToF-AMS) provided non-refractory PM$_1$ composition and elemental ratios. AMS-PMF was applied to further categorize PM$_1$ OA. PM$_{2.5}$ filter samples were collected on a day-night basis and were analyzed for OC, ionic species, organic molecular markers, and SOA tracers from biogenic and anthropogenic precursors. CMB and MM-PMF modeling were applied to apportion the primary and secondary sources
of PM$_{2.5}$ OC. The outcomes of these source apportionment models were compared and are discussed along with the meteorological data and real time measurements of VOCs in order to gain a robust understanding for the sources, abundance, and variability of fine PM, particularly SOA, in HSC.

## 2 Experimental methods

### 2.1 Site description

Fine PM was studied at the Clinton Drive monitoring site in Houston, TX (29.733943° N, 95.257684° W), that is maintained by the Texas Commission on Environmental Quality (TCEQ). Clinton Drive is located 11 km west of the city center and is adjacent to the HSC. The immediate surroundings include industrial facilities (e.g., oil refineries), heavily trafficked roadways, and several neighborhoods.



## 2.2 Co-located measurements

The Clinton Drive site provided access to co-located hourly measurements of VOC that were measured using an automated gas chromatograph (GC), fine PM (PM$_{2.5}$) measured by tapered element oscillating microbalance (TEOM), and meteorology. Quality-controlled data were collected,  and accessed through TCEQ (TCEQ, 2017).

## 2.3 High resolution time of flight aerosol mass spectrometer (HR-ToF-AMS)

### 2.3.1 PM$_1$ measurements

Real time measurements of NR-PM$_1$ were taken with a HR-ToF-AMS (Aerodyne Research) for the period 13-29 May, 2015 at the Clinton Drive monitoring site, with a time resolution of 1 minute.  The HR-ToF-AMS has been described in detail previously (DeCarlo et al., 2006); the sampling protocol utilized in this study is identical to that described by Wallace et al. (2018).

### 2.3.2 Positive matrix factorization analysis of PM$_1$ OA (AMS-PMF)

Source apportionment of the organic fraction of NR-PM$_1$ was conducted using AMS-PMF (Paatero and Tapper, 1994). The PMF evaluation tool (PET v.2.08D, (Ulbrich et al., 2009)) employing the PMF2 algorithm (Paatero, 2013) running in robust mode with model error set to 0 was used for analysis of the HR OA mass spectra ($m/z$=12 to $m/z$= 115). The HR OA concentration matrix and associated error matrix resulting from PIKA v 1.16H were used as input for AMS-PMF application. Prior to analysis, mass fragments with signal to noise ratio (SNR) below 0.2 were removed from the dataset, while ions with SNR between 0.2 and 2 were down weighted by a factor of 3, following Paatero and Hopke (2003). Similarly, a down weighting factor of $\sqrt{5}$ was applied to CO$_2^+$-related ions to prevent excessive influence of the m/z 44 signal, as recommended by Ulbrich et al. (2009). AMS-PMF model solutions including from 1 to 7 factors were fit using multiple initialization points in order to ensure convergence to global rather than local minima. Each set of AMS-PMF model output was evaluated based on the ratio of the summation of the scaled residuals (Q) to the expected value of Q (Q/Qexpected) (Supplemental Information; Table S1 and Fig. S1-S3), its convergence to a global minimum, and its ability to reproduce the OA mass concentrations measured during the field campaign. These parameters were examined and used as selection criteria for the number of factors in the final AMS-PMF model. Additionally, the physical meaningfulness of the retained factors in each model and their similarity with factors reported in previous OA studies employing HR-ToF-AMS were considered when selecting the number of AMS-PMF components. Based on these criteria, a model including three factors was found to be the most appropriate for describing the dataset under analysis. Further details on AMS-PMF application and the criteria for factor selection are presented in Fig. S1-S3.

## 2.4 Filter sample collection and offline chemical analysis

### 2.4.1 Filter sample collection

PM$_{2.5}$ samples were collected using a medium-volume URG air sampler (3000B, URG Corp.) with a cyclone (URG) operating at a flow rate of 90 L min$^{-1}$. Air flow rate was monitored before and after sampling using a rotameter (Gilmont Inst.). PM$_{2.5}$ samples were collected on 90-mm quartz fiber filters (Pallflex® Tissuquartz™, Pall life science) that were pre-cleaned by baking for 18 hours at 550 °C. Samples were collected for the period 5-27 May 2015 twice daily, during daytime (7:00 - 18:00 LT) and nighttime (19:00 – 6:00 LT). After sampling, filters were transferred to Petri dishes lined with pre-baked aluminum foil, sealed with Teflon tape, transported to the laboratory, and stored frozen at -20 °C until analysis.





One field blank was collected for every five samples by loading a blank filter into the filter holder, pulling no air through it, and removing it from the filter holder.

### 2.4.2 Measurements of organic carbon, elemental carbon, and organic species

Organic carbon (OC) and elemental carbon (EC) were measured by thermal-optical analysis (Sunset Laboratory Inc.) on a 1 cm$^2$ filter portion following Schauer et al (2003). Filters were extracted into acetonitrile following the method described by Al-Naiema and Stone (2017). Briefly, isotopically-labelled internal standards were added onto each filter. Then, filters were extracted sequentially with three 10 mL portions of acetonitrile (Optima-Fisher Scientific-Fisher Chemical) for 15 minutes by ultra-sonication (Branson 5510, 60 Sonics per minute). The combined extracts were reduced to 2 mL by rotary-evaporation at 30 °C, 120 rpm, and 200 mbar (Heidolph, Hei-vap G1). Extracts were filtered with 0.25 μm PTFE syringe filters (Whatman) and stored frozen at – 20 °C. Immediately prior to analysis, the extracts were evaporated to 100 μL under a gentle stream of ultra-pure nitrogen at 30 °C. All glassware used in extraction was first baked (500 °C for 5 hours) to remove organic contaminants and then silanized using 5% solution of dichlorodimethylsilane (Fluka), prepared in toluene (Sigma-Aldrich). Organic species were analyzed using an Agilent 7890A GC coupled to a 5975C MS (Agilent Technologies). Polycyclic aromatic hydrocarbons (PAH), n-alkanes, and hopanes were directly injected to the GC-MS equipped with a DB-5 column and electron impact (EI) ionization source (70 eV). The GC inlet temperature was 300 °C. An aliquot of the extract was trimethylsilylated with N,O-bis(trimethylsilyl)trifluoroacetamide with trimethylchlorosilane (BSTFA+TMCS, 99:1, Fluka Analytical 99%). A 20 μL aliquot of the extract was dried under a gentle stream of nitrogen, 10 μL of the silylation agent was added, and the mixture was reacted at 100 °C for 90 min. Details about species quantification by GC-MS are provided elsewhere (Al-Naiema and Stone, 2017).

### 2.4.3 Ion Analysis and pH estimation

Filters were extracted into ultra-pure (UP) water (Barnstead EasyPure II) by shaking (125 rpm) for 10 minutes, sonication (60 sonics min$^{-1}$) for 30 minutes, followed by shaking (125 rpm) for 10 minutes. Extracts were filtered with 0.45 μm syringe filters (PTFE, Whatman). Ions were analyzed by a Dionex ICS-5000 ion chromatograph (Dionex ASDV). Details regarding ion separation and quantification are described elsewhere (Jayarathne et al., 2016).

Ion results along with other meteorological data such as relative humidity and ambient temperature were introduced to the Extended Aerosol Inorganics (E-AIM IV) model (Friese and Ebel, 2010), available interactively from http://www.aim.env.uea.ac.uk/aim/model4/model4a.php, to estimate aerosol pH. In this study, model input included the molar concentrations of sulfate, nitrate, chloride, ammonium, sodium, calcium, and magnesium; the pH was estimated by calculating the [H$^+$] required to balance any cation deficiency.

### 2.5 Chemical mass balance (CMB) modeling

The contribution of different sources to the OC fraction was estimated using the EPA CMB receptor model (v8.2). CMB model employs source profiles to estimate source contributions to ambient PM by solving for the least-squares solution (Watson et al., 1984). In this study, the input source profiles included diesel engines and both smoker and non-smoker gasoline engines (Lough et al., 2007); secondary organic carbon (SOC) from isoprene, α-pinene, and toluene (Kleindienst et al., 2007); bituminous coal (Oros and Simoneit, 2000); biomass burning (Lee et al., 2005); and ship emissions (Agrawal et al., 2010). The naphthalene SOA profile was obtained from Kleindienst et al. (2012) and had a phthalic acid-to-SOC mass fraction of 0.0389, phthalic acid-to-SOA mass fraction of 0.0199, and the average SOA:SOC of 1.95. Species included in the CMB model included 17α(H)-21β(H)-hopane, 17α(H)-22,29,30-trisnorhopane, 17β(H)-21α(H)-



30-norhopane, ABB-20(R+S)-C27-cholestane, ABB-20(R+S)-C29-sitostane, n-alkanes ($C_{25}$-$C_{31}$), PAH (benzo(b)fluoranthene, benzo(k)fluoranthene, benzo(e)pyrene, indeno(1,2,3-cd)pyrene, benzo(ghi)perylene, and dibenz(ah)anthracene), isoprene SOA tracers (2-methylglyceric acid and 2-methyltetrols), one α-pinene SOA tracer (cis-pinonic acid), one naphthalene SOA tracer (phthalic acid), and one toluene SOA tracer (2,3-dihydroxy-4-oxopentanoic acid).

**2.6 Molecular marker based-positive matrix factorization (MM-PMF)**

The EPA PMF (version 5) was used for source apportionment of $PM_{2.5}$ OC based on organic species and EC as input data. The MM-PMF input data statistics are summarized in Table S2. MM-PMF solutions for 3 to 11 factors were analyzed using 20 base runs. To determine a final solution, MM-PMF solutions with 5 to 9 factors were further analyzed using 100 base runs each starting with a random seed. The stability of the PMF solutions were assessed using displacement (DISP), bootstrapping (BS), and BS-DISP error estimation methods following the recommendations of Brown et al. (2015) and Norris et al. (2014). PMF settings for base runs and error estimation are summarized in Table S3.

**2.7 Statistical analysis**

Correlation analysis among the measured species and VOCs were evaluated using Minitab statistical analysis software (version 17). Correlations were interpreted as follows: very high (0.9-1.0), high (0.7-0.9), moderate (0.5-0.7), low (0.3-0.5), and negligible (0.0-0.3) (Mukaka, 2012). The statistical significance of correlations was evaluated at the 95% confidence interval ($p < 0.05$).

**3 Results and discussion**

Clinton Drive is a long-term monitoring site near the HSC in Houston where $PM_{2.5}$ mass, select gases, and meteorology are measured hourly (TCEQ, 2017), Fig. S4. Several extreme rain events and flooding occurred during the study period of 5-27 May 2015 (Fig. S5). Winds were predominantly southerly, transporting air from the Gulf of Mexico, suggesting minimal influence of continental transport on ambient air at Clinton Drive. Daily $PM_{2.5}$ mass concentrations averaged $14.0 \pm 5.1$ µg m$^{-3}$ and ranged from 4.4 to 30.8 µg m$^{-3}$, well below the daily National Ambient Air Quality Standard (NAAQS) of 35 µg m$^{-3}$ (US-EPA). Hourly $PM_{2.5}$ concentrations peaked between 7-10 am, coinciding with morning traffic. The $NO_x$ and toluene mixing ratios also peaked in the early morning and late afternoon, coinciding with high traffic periods (Fig. S4). The daytime peaks in ozone ($O_3$) and isoprene were consistent with expected summertime trends.

**3.1 Overview of non-refractory submicron aerosol composition measured by HR-ToF-AMS**

Non-refractory (NR) $PM_1$ measurements by the HR-ToF-AMS from 13 to 29 May are summarized in Fig. 1. Due to the rainy weather, the observed NR-$PM_1$ levels were low in comparison to other measurements made in the HSC area earlier in 2015 (Wallace et al., 2018). Despite the rainy conditions, periods of elevated NR-$PM_1$ loadings occurred (Fig. 1a). Average relative contributions from major species to NR-$PM_1$ quantified are shown in Fig. 1b. Sulfate and organics are the two most abundant components of the NR-$PM_1$, contributing 44.9% and 39.7%, respectively. Sulfate and organics exhibited periods of high loadings with concentrations over 15 µg m$^{-3}$ and maximum 1-minute averaged concentrations of 22.2 and 57.5 µg m$^{-3}$, respectively. Ammonium was the next most abundant species, making up 13.9% of the NR-$PM_1$, followed by nitrate and chloride, which only contributed trace amounts (1% or less) to the NR-$PM_1$ mass concentration. Table 1 summarizes the mean, median, standard deviation, and range of 1-minute concentrations for NR-$PM_1$ species. The diurnal profiles for $PM_1$ species (Fig. 1c) indicate no diurnal trend for the measured species, except for organic aerosol, which





exhibited higher concentrations during daytime. The elemental ratios of the organic portion of NR-PM$_1$, including O:C and H:C, as well as OM:OC and the average oxidation state ($\overline{OSc}$) are presented in Table 2.

### 3.2 Composition of PM$_{2.5}$ determined by filter-based measurements

Filter-based PM$_{2.5}$ measurements indicated that, on average, organic carbon (OC) and elemental carbon (EC) contributed 17% and 4% of PM$_{2.5}$, respectively (Table 3). Organic matter (OM) was estimated by the mean OM:OC ratio of 2.11 measured by HR-ToF-AMS (Table 2 and Fig. S6) to contribute 35% of PM$_{2.5}$. The AMS-determined OM:OC ratio is considered to be the best estimate of the PM$_{2.5}$ OM:OC ratio, since it was determined at HSC for the study period. However, this estimation is limited by the differences of sizes of particles analyzed by each method as well as AMS's measurement of only non-refractory OA, while PM$_{2.5}$ OC includes refractory and non-refractory species. OM concentrations were significantly higher during daytime ($4.8 \pm 1.2$ µg m$^{-3}$) compared to nighttime ($3.6 \pm 1.4$ µg m$^{-3}$, p = 0.011, Table 3). EC concentrations were also significantly higher during daytime (averaging $0.7 \pm 0.4$ µg m$^{-3}$) compared to nighttime ($0.3 \pm 0.2$ µg m$^{-3}$, p < 0.001). Higher daytime EC at Clinton Drive is expected to be influenced by transportation emissions near the HSC (Levy et al., 2013; Zhang et al., 2017). On average, sulfate contributed 21.4% of PM$_{2.5}$ (averaging $2.87 \pm 1.39$ µg m$^{-3}$), with minor contributions from ammonium (4%, $0.52 \pm 0.40$ µg m$^{-3}$), nitrate (3%, $0.40 \pm 0.37$ µg m$^{-3}$), sodium (5.5%), chloride (3.3%), potassium (0.6%), and magnesium (0.7%). Calcium contributed 4.3% of PM$_{2.5}$ and likely originated from road dust, which has previously been estimated to contribute to 11% of PM$_{2.5}$ at Clinton Drive (Fraser et al., 2003). PM$_{2.5}$ mass not accounted for by the measured species (Fig. 2) is expected to arise from unmeasured species such as crustal metal oxides (e.g. silica, alumina), other metals, and particle-bound water. For samples in which the measured species exceed PM$_{2.5}$ mass, contributing factors include analytical uncertainties in chemical species measurements and PM$_{2.5}$ mass measured by tapered element oscillating microbalance (TEOM) (Ayers et al., 1999). Filter-based PM$_{2.5}$ measurements indicate the same major PM species as AMS NR-PM$_1$ measurements and also capture refractory PM components (i.e., EC, a fraction of OM, and crustal elements).

Ion measurements indicate that the aerosol at Clinton Drive is acidic. The correlation between the molar concentrations of major anions (sulfate and nitrate) and ammonium measured by HR-ToF-AMS (Fig. 3) had a slope of $1.287 \pm 0.002$, indicating that ammonium does not fully neutralize sulfate as ammonium sulfate. Aerosol pH was estimated using E-AIM IV to range from 0.29 to 1.45 with an average of $0.44 \pm 0.39$. The estimated pH values might be biased because E-AIM does not account for the activity coefficient of H$^+$ and neglects the role of organic acid dissociation (Hennigan et al., 2015). In comparison to other locations in the summertime, the estimated pH values in HSC are less than those estimated by ISORROPIA-II for Birmingham, Alabama (1.6 – 1.9) (Rattanavaraha et al., 2016) and Centreville, Alabama (0.5 - 2) (Guo et al., 2015), but are higher than those estimated by E-AIM for four major cities in China (-0.77 to -0.52) (Pathak et al., 2009). Acidic aerosol is expected to enhance SOA formation, as indicated in previous SOA chamber experiments (Surratt et al., 2007; Iinuma et al., 2004).

### 3.3 PMF of AMS data: Factor identity and contribution to OA

The mass spectra of the three-factor AMS-PMF solution are presented in Fig. 4. Each OA factor exhibited marked differences in their spectral mass signature and fragmentation patterns. Elemental ratios of O:C and H:C for the factors ranged from 0.06 to 1.24 and 1.21 to 2.03, respectively, while $\overline{OS}$c ranged from -1.91 to 1.27 (Table S4). These metrics reflect a largely different chemical character of the retained factors and indicate the likely contribution of components with primary and secondary origin to the observed OA concentrations (Zhang et al., 2011).



The hydrocarbon-like organic aerosol (HOA) factor in Fig. 4 had large signals at *m/z* 41, 43, 55 and 57 and significant contributions from mass fragments above *m/z* 60. These characteristics and O:C and H:C ratios are typical of primary organic aerosol (Zhang et al., 2011; Aiken et al., 2008). This classification was further confirmed by spectral contrast angles (θ) of ~13 to 15° between this factor and previously reported HOA factors (Aiken et al., 2009; Docherty et
al., 2011; Mohr et al., 2012; Elser et al., 2016), indicating strong similarities in the mass spectra  HOA (Kaltsonoudis et al., 2017).

The low volatility oxygenated organic aerosol (LV-OOA) and cooking-influenced semi-volatile oxygenated organic aerosol (CI-SV-OOA) factors were characterized by large fractions of *m/z* 44 ($f_{44}$, associated mostly to $CO_2^+$ signal) and by elevated O:C ratios and $\overline{OS}c$ levels (Table S4), which are indicative of atmospherically processed OA aerosol with a likely
secondary origin (Zhang et al., 2011). Moreover, $f_{44}$ and the fraction of *m/z* 43 in the mass spectrum ($f_{43}$, mainly related to the $C_2H_3O^+$ ion; Table S4) locate them in the SV-OOA and LV-OOA regions of the "triangle plot," respectively as introduced by  Ng et al. (2011).

The mass spectrum of the CI-SV-OOA factor (Fig. 4) closely coincides with at least one SV-OOA factor included in the UCB-AMS Database (θ ~13 °) (Aiken et al., 2009) and exhibits statistically significant (p<0.01) linear association
with semi-volatile $PM_1$ constituents such as nitrate (r=0.6). Further insight on the identity of this factor was obtained by examining its correlation with markers of OA sources; this analysis showed a significant correlation of the time series of the SV-OOA factor with mass fragments previously reported as tracers of cooking organic aerosol (COA) such as $C_3H_3O^+$, $C_3H_5O^+$, $C_2H_3O^+$ and $C_5H_8O^+$ (r= 0.7-0.9, p<0.01) (Mohr et al., 2012; Mohr et al., 2009; Sun et al., 2011).

The LV-OOA factor (Fig. 4) exhibited a statistically significant moderate correlation with particulate sulfate levels
(r=0.5, p<0.01), suggesting, to some extent, its formation on a regional scale (Peng et al., 2016).  These observations along with the resemblance of the mass signature of this factor with LV-OOA factors reported in previous studies (θ below 17°) (Docherty et al., 2011; Mohr et al., 2012), led to the classification of this factor as atmospherically processed OA resembling LV-OOA.

The time series of concentration of the HOA, CI-SV-OOA, and LV-OOA factors and their diurnal trends are
presented in Fig. 5. The average mass concentrations observed for HOA, CI-SV-OOA, and LV-OOA during the field campaign were 0.72 ±0.52 μg m$^{-3}$, 0.48 ± 0.47 μg m$^{-3}$, and 0.37 ± 0.73 μg m$^{-3}$, respectively, indicating a predominant contribution from secondary factors to the $PM_1$ OA. The contribution to the OA mass concentration during the sampling period followed the sequence LV-OOA > CI-SV-OOA > HOA, with average abundances of approximately 48.23, 29.44, and 22.33%, respectively.

As indicated by the standard deviations associated with the average mass concentrations of the PMF factors (Figure 5), large variations in their OC contributions occurred during the sampling interval, with particularly high variability observed for HOA. As presented in Fig. 5b, the diurnal profile of the HOA factor exhibited local maxima at ~09:00 and 17:00 LT, respectively, indicating its enhancement during periods of significant traffic activity. Thus, it is likely that the observed variability in the mass concentration of HOA is related peak traffic times. According to Fig. 5b, LV-OOA mass
concentrations showed a relatively flat diurnal behavior with a slight increasing trend during daytime  (7:00 to 18:00 LT), which is consistent with periods of enhanced photochemical activity and is in agreement with LV-OOA diurnal patterns reported in previous studies (Sun et al., 2011, Zhang et al., 2016). The hourly behavior of the CI-SV-OOA factor (Fig. 5b) indicates higher concentrations during daytime, unlike the expected trends for semi-volatile species. Although a slight concentration peak at ~12:00 LT, likely related with cooking activities, was noticed in this factor, no evident late-night
increases as those previously observed for COA in the Houston area (Wallace et al., 2018) were observed (Fig. 5b).




### 3.4 Source apportionment of PM$_{2.5}$ OC in HSC

#### 3.4.1 Chemical mass balance (CMB) modeling

CMB modeling apportioned PM$_{2.5}$ OC to eight sources (Fig. 6, Table 4): diesel engines, gasoline engines (reported as the sum of smoking and non-smoking gasoline engines), biomass burning (BB), ship emissions, isoprene SOA, α-pinene SOA, monoaromatic SOA, and naphthalene SOA. Unapportioned (or other) OC was calculated as the difference between the observed OC mass and the OC apportioned to these eight sources. The average OC during daytime (2.27 ± 0.56 μgC m$^{-3}$) was apportioned 55% to primary sources and 16% to secondary sources, with 29% unapportioned. The average OC mass during nighttime was apportioned 37% to primary sources and 14% to secondary sources, with 49% unapportioned.

Motor vehicles were the greatest PM$_{2.5}$ OC source, with gasoline engines contributing 30% and diesel engines contributing 10% on average. OC contributions from gasoline engines were significantly higher during daytime (0.82 ± 0.37 μgC m$^{-3}$) compared to nighttime (0.36 ± 0.30 μgC m$^{-3}$, p<0.001, Table 4). Similarly, diesel engine contributions were significantly higher (p=0.001) during daytime (0.27 ± 0.15 μgC m$^{-3}$) compared to nighttime (0.13 ± 0.09 μgC m$^{-3}$). The higher daytime contributions are expected to result from greater motor vehicle activity during daytime, which captured the majority of peak traffic times in the morning and afternoon.

Biomass burning had a small impact on PM$_{2.5}$ OC, with an average contribution of 5% (0.10 μgC m$^{-3}$). No significant differences in daytime and nighttime concentrations were observed. The open BB profile was used in CMB because high fire activity was observed in the Yucatan Peninsula of Mexico during the time of sample collection (Fig. S7). Backward wind trajectories indicated that some air masses affecting Houston had travelled over the Yucatan Peninsula (Fig. S8). The influence of Mexico wildfire on the Houston airshed, previously noted in other studies, typically peaks during the month of May (Duncan et al., 2003; Yokelson et al., 2009; Crounse et al., 2009). A similarly minor contribution from BB was previously reported for the same sampling site (Fraser et al., 2003).

Ship emissions contributed 1% of PM$_{2.5}$ OC, with a significantly higher (p=0.001) daytime concentration (0.02 ± 0.01 μgC m$^{-3}$) compared to nighttime (0.01 ± 0.01 μgC m$^{-3}$; Table 4). Despite the location of Clinton Drive near the HSC, ship emissions were not a major source of OC, which may be due to the prevailing wind direction (Fig. S4). These results may be also biased from the use of a single source profile in CMB modeling, since ship characteristics such as vessel category, speed, and loading impact ship emissions (Williams et al., 2009).

CMB was used to apportion OC to four SOA precursors, following the SOA-tracer approach (Kleindienst et al., 2007). SOA from monoaromatic VOCs was estimated by 2,3-dihydroxy-4-oxopentanoic acid (DHOPA, detected in 80% of samples) at 3% of OC (Table 4). Toluene is a known precursor to DHOPA (Kleindienst et al., 2004) and monoaromatic SOA correlated significantly with toluene during daytime (r=0.52, p=0.039) and nighttime (r=0.725, p<0.001). The diurnal trend in toluene concentrations coincides with peak traffic times (Fig. S4) suggesting that vehicles are the major source of monoaromatic SOA precursors.

Naphthalene SOA contributed an average of 4.6% to PM$_{2.5}$ OC (Table 4). To our knowledge, this is the first study to use ambient concentrations of phthalic acid in CMB modeling to estimate naphthalene SOA, following our previous recommendations (Al-Naiema and Stone, 2017). Phthalic acid concentrations were converted to SOA yields using the mass fraction of phthalic acid-to-OC in SOA generated in naphthalene photooxidation chamber experiments (Kleindienst et al., 2012). The estimated naphthalene SOA correlated significantly with gasoline engines (r = 0.409, p = 0.012), suggesting gasoline engines are an important sources of naphthalene, which is consistent with a previous report of naphthalene accounting for 56% of PAH emitted from gasoline engines (Khalili et al., 1995).

Isoprene SOA was estimated to contribute 7% of the PM$_{2.5}$ OC by way of three tracers: 2-methylthreitol, 2-methylerythritol, and 2-methylglyceric acid (Fig. 6, Table 4). On average, isoprene SOA was higher during daytime (0.18 ± 0.11 μgC m$^{-3}$) compared to nighttime (0.11 ± 0.13 μgC m$^{-3}$), consistent with the hourly diurnal profile of isoprene that



follows ambient temperature and UV radiation (Fig. S4) and prior studies of isoprene SOA (Budisulistiorini et al., 2015; Xu et al., 2015). Industrial emissions can be sources of isoprene in HSC, with two point sources identified near the sampling location (Fig. S9). Isoprene SOA contributions were seven times lower than those reported in June-July 2013 in Look, Rock TN, which has a much higher isoprene concentration (2 ppbv) (Budisulistiorini et al., 2015) than the HSC area average during this study (0.1 ppbv,Fig. S4).

α-Pinene SOA contributed 0.5% of $PM_{2.5}$ OC, based on the ambient concentrations of cis-pinonic acid that forms by ozonolysis of α-pinene (Christoffersen et al., 1998). Only one α-pinene SOA tracer was detected, consistent with the low mass contributions of this source to OC and the low monoterpene emission potential near the HSC (Brown et al., 2013).

The CMB-source apportionment of primary sources (contributing an average of 46% of OC) agrees well with previous studies in the HSC. For a non-wood smoke event of summer 2000, Buzcu et al. (2006) reported that primary sources contributed 49% to OC, with contributions from diesel (21%) and gasoline (15%) vehicles, BB (8%) and meat cooking (3%). Absolute contributions of these primary sources to OC have decreased by 33-83% over that last decade, when comparing this study to Buzcu et al., (2006) In February 2015, Wallace et al. (2018) identified three primary $PM_1$ OA factors: hydrocarbon-like (14%), BB (22%), and cooking (8%). Altogether, this and prior studies indicate that motor vehicles contribute significantly to OC year-round, that summertime contributions from BB are smaller than winter, and that cooking contributions are relatively small. The SOA-tracer method was applied to the HSC for the first time, and yielded the estimates that anthropogenic SOA from monoaromatic VOC and naphthalene contributed an average of 7.5% to $PM_{2.5}$ OC, with biogenic SOA from isoprene and α-pinene contributing 7.4% of OC. Notably, a major fraction of OC was unapportioned, averaging 29% in the daytime samples and 49% in the nighttime samples. Considering the strong agreement of primary source contributions with the other two approaches in this work (section 3.6) and in prior studies, it is expected that the unapportioned OC is due to SOA. The SOA not accounted for in the CMB model includes SOA precursors for which the SOA tracer method has not been developed and also arises from differences in the SOA tracer-to-OC ratios across chamber experiments and the Houston airshed.

### 3.4.2 Molecular marker-based positive matrix factorization model (MM-PMF)

The 9 factor solution was identified as the optimal solution by analyzing Q, error estimation diagnostics and factor interpretability (Table S5). The difference between $Q_{robust}$ and $Q_{true}$ is smallest for the 9-factor solution, indicating a minimum impact from outliers. The difference between $Q_{robust}$ to $Q_{expected}$ ratio is smallest when moving from 8 to 9 factors in the solution. The base model diagnostics and error estimation for the 9-factor solution are summarized in Table S2and Fig. S10, respectively. The sources associated with each factor were identified by the key chemical species apportioned to each factor (Fig. 7), factor contributions (Fig. 8), and factor correlations with co-located measurements and CMB source contribution estimates.

The diesel engines factor contributed 12% of average OC (equivalent to 0.22 µgC m$^{-3}$). The key chemical species apportioned to this factor include cyclopenta(cd)pyrene (76%), benz(a)anthracene (88%), chrysene (64%), EC (41%), and 17β(H)-21α(H)-norhopane (39%) that are components of fossil fuel combustion emissions (Lough et al., 2007; Rogge et al., 1993a). This factor contributes significantly more OC during daytime (0.369 µgC m$^{-3}$) compared to nighttime (0.066 µgC m$^{-3}$, p<0.001). The factor EC:OC ratio of 0.92 suggests contributions from light-duty (<33,000 lb) diesel-powered motor vehicles (1.9±0.53) (Lough et al., 2007). The factor identification is further supported by its positive correlation with the CMB-diesel engine source (r=0.727, p<0.001) and slope of 1.5±0.2.

The gasoline engines factor contributed an average of 24% of OC (0.46 µgC m$^{-3}$). The major chemical species apportioned to this factor include n-alkanes such as tetracosane (37%), pentacosane (39%), hexacosane (38%), heptacosane (33%), octacosane (50%), and nonacosane (37%), and 17β(H)-21α(H)-norhopane (38%) that have been detected among fossil fuel combustion emissions (Lough et al., 2007; Rogge et al., 1993a). The EC:OC ratio of this factor (0.29) is within the





range of EC:OC ratios for non-smoking (0.20 to 0.52) and smoking (0.0 to 2.5) gasoline vehicles (Lough et al., 2007). The moderate and significant correlation of this factor with CMB-gasoline engines (r=0.479, p=0.001) and slope of 0.6±0.2 further support the identification of this factor as gasoline engines.

The non-tailpipe vehicle emissions factor contributed an average of 11% of $PM_{2.5}$ OC (0.21 µgC m$^{-3}$). The key chemical species apportioned to this factor includes 17α(H)-21β(H)-hopane (82%), pristane (32%), and nonadecane (20%), while the absence of EC indicates a non-combustion source. 17α(H)-21β(H)-Hopane is a tracer for fossil fuel combustion and has been detected in both tailpipe and non-tailpipe vehicle emissions and is present in the higher boiling point fractions of crude oil that are used to manufacture lubricating oils, waxes, tires, and asphalt (Rogge et al., 1993a, b). Hopanes in the atmosphere come from engine oil evaporation, tire wear, and paved road dust and to a lesser extent from brake wear particles (Rogge et al., 1993a, b). Pristane and nonadecane have also been detected in tire dust, brake lining wear particles, and paved road dust particles (Rogge et al., 1993b). This factor contributes significantly more OC during daytime (0.249 µgC m$^{-3}$) compared to nighttime (0.171 µgC m$^{-3}$, p=0.050) (Fig. 8). This factor identification is further supported by the ratio of benzo(a)pyrene to the sum of benzo(a)pyrene and chrysene of this factor (0.29), which is within the range of non-tailpipe vehicle emissions (0.23-0.32) (Rogge et al., 1993b). The factor contribution averaged 29% of OC for samples in which 17α(H)-21β(H)-hopane was detected, suggesting that the detectability of this tracer influenced this factor's contributions to OC.

The ship emissions factor contributed an average of 2% of $PM_{2.5}$ OC (0.04 µgC m$^{-3}$). The key species attributed to this factor were benzo(b)fluoranthene (42%), benzo(k)fluoranthene (79%), benzo(e)pyrene (40%), benzo(a)pyrene (77%), indeno(1,2,3-cd)pyrene (42%), and benzo(ghi)perylene (51%), nonadecane (34%), 5-nitro-salicylic acid (36%), and 2-methyl-4-nitrophenol (59%). The PAH and *n*-alkanes indicate a primary fossil fuel combustion source, while the nitromonoaromatic compounds can be either emitted by fossil fuel combustion or formed by the photooxidation of aromatic VOC in the presence of $NO_x$ (Al-Naiema and Stone, 2017; Harrison et al., 2005; Lin et al., 2015). Since these nitromonoaromatic compounds are primarily attributed to this factor, fossil fuel combustion is expected to have been their major source. The absence of EC in this factor is consistent with the very small EC:OC ratio of ship emissions (0.03±0.002) and the ratio of indeno(1,2,3-cd)pyrene to the sum of indeno(1,2,3-cd)pyrene and benzo(ghi)perylene of 0.28 that is similar to that of ship emissions (0.36) (Agrawal et al., 2010).

The cooking factor contributed an average of 1% of $PM_{2.5}$ OC (0.02 µgC m$^{-3}$). The key chemical species apportioned to this factor include cholesterol (90%), a tracer for meat cooking (Rogge et al., 1991), 4-methyl-3-nitrophenol (86%) and *n*-alkanes such as docosane (51%), tricosane (40%), and heptacosane (34%) that have been also been detected during commercial food cooking (Roe et al., 2004; Rogge et al., 1991). Factor contributions were observed only in samples with detectable levels of cholesterol, for which the average contribution to OC was 2.4%. Overall, these results suggest cooking was a minor source of $PM_{2.5}$ OC in this study.

The BB factor contributed 11% of OC on average (0.20 µgC m$^{-3}$). This factor is the major source of levoglucosan (60%), a tracer for BB emissions (Simoneit et al., 1999). Other key species apportioned to this factor include SOA products such as isophthalic acid (38%) and cis-pinonic acid (63%) that have been observed among aged BB emissions (Yan et al., 2008). The factors' EC:OC ratio (0.044) is also closer to the EC:OC ratio of aged BB emissions (0.039) that contain both primary and secondary BB aerosols (Yan et al., 2008) compared to fresh, primary BB emissions (0.065) (Lee et al., 2005). These results suggest that the BB emissions observed in the sampling site represent aged BB emissions that were likely transported from Yucatan Peninsula in Mexico (Fig. S7, S8). These results are consistent with prior studies that reported BB emissions in southern Texas were transported from Mexico during the months of April to May during major fire events (Kaulfus et al., 2017; Rogers and Bowman, 2001; Wang et al., 2006).

The factor identified as isoprene SOA contributed an average of 11% of OC (0.20 µgC m$^{-3}$). The key species apportioned to this factor include 2-methylthreitol (56%) and 2-methylerythritol (62%) which are isoprene SOA tracers from the photooxidation of isoprene under low-$NO_x$ conditions (Lin et al., 2013). This factor is significantly higher during





daytime (0.275 µgC m$^{-3}$) than nighttime (0.134 µgC m$^{-3}$; p=0.036) which is consistent with the high photochemical activity and high isoprene emissions that is triggered by high daytime temperatures and sunlight (Sharkey et al., 1996). This factor identification is further supported by its very high and positive correlation with CMB-isoprene SOA (r=0.934, p<0.001), although the slope of 1.8 ± 0.1 suggests the SOA tracer method in CMB underestimated its contribution to OC.

The high-NO$_x$ anthropogenic SOA factor contributed 7% of OC on average (0.12 µgC m$^{-3}$). The key chemical species apportioned to this factor include 4-methyl-2-nitrophenol (74%), DHOPA (48%) and 2-methylglyceric acid (54%). 4-Methyl-2-nitrophenol forms by the photooxidation of monoaromatic compounds such as toluene, *p*-xylene and *p*-ethyltoluene in the presence of NO$_x$ (Forstner et al., 1997). Similarly, DHOPA is formed by photooxidation of toluene under high-NO$_x$ conditions (Kleindienst et al., 2007). 2-Methylglyceric acid forms from methacrolein (MACR) in the presence of
NO$_x$; MACR can form by the oxidation of isoprene (Nguyen et al., 2015) or can be directly emitted from vehicles (Park et al., 2011), making it either biogenic or anthropogenic, respectively. Vehicle emissions are expected to be the major source of MACR in an urban site located close to the HSC (Park et al., 2011), suggesting that 2-methylglyceric acid in HSC is likely to originate from anthropogenic sources. This factor identification is further supported by its high positive correlation with CMB-monoaromatic SOA (r=0.754, p<0.001).

The factor identified as low-NO$_x$ anthropogenic SOA contributed 22% of OC on average (0.41 µgC m$^{-3}$). The key species apportioned to this factor include phthalic acid (38%), 4-methylphthalic acid (43%), terephthalic acid (42%), and 4-nitrophenol (36%). Phthalic acid and 4-methylphthalic acid are recommended as SOA tracers for naphthalene and methylnaphthalene, respectively (Al-Naiema and Stone, 2017). Nitrophenols have also been detected during the photooxidation of PAH (Kautzman et al., 2010). In addition, this factor moderately and positively correlates with CMB-
naphthalene SOA (r=0.510, p<0.001). The SOA tracer method estimate of naphthalene SOA in CMB is only 22% of this factor, suggesting that the SOA tracer method underestimates the extent of low-NO$_x$ anthropogenic SOA based on phthalic acid concentrations alone and/or that this factor contains SOA from other VOC precursors. In addition to naphthalene, several laboratory studies have shown that *n*-alkanes, lighter aromatics and other PAH, which are mainly emitted from fossil fuel combustion and industries, also contribute to SOA (Chan et al., 2009; Gentner et al., 2012; Zhang and Ying, 2012).
Further identification and quantification of anthropogenic SOA tracers from other VOC precursors would improve the ability of the MM-PMF to more broadly capture the magnitude of anthropogenic SOA.

**3.5 Source apportionment of fine organic aerosol in HSC: a three-method approach**

Herein, source apportionment results obtained from AMS-PMF for PM$_1$ OC (converted from OA using OM:OC ratios in Fig. 4), and PM$_{2.5}$ OC by MM-PMF and CMB models are integrated (Table 5 and Fig. 9) with consideration of the
strengths and weaknesses of each approach. The contribution from primary fossil sources are defined as the sum of contributions of diesel engines, gasoline engines, and ship emissions from CMB (41%), the sum of the three aforementioned sources with non-tailpipe vehicle emissions from MM-PMF (49%), and HOA factor from AMS-PMF (37%). A similar contribution from primary fossil sources to PM$_{2.5}$ OC (36%) was reported by Buzcu et al. (2006) in Houston near the HSC area during a non-wood smoke event. HOA resolved in AMS-PMF highly correlated with OC from diesel engines in MM-
PMF (r=0.824, p<0.001) and CMB (r=0.890, p<0.001), and moderately correlated with OC from gasoline engines (r=0.645, p<0.001) and ship emissions (r=0.696, p<0.001) in CMB. These correlations indicate temporal consistency among the fossil sources of OC and the percent contributions to OC indicate that the three models resolved a consistent fossil contribution to organic aerosol. Further, the models agree that motor vehicle emissions are the major contributor to fossil-fuel derived organic aerosol, making them the dominant primary PM source in HSC. The agreement of the three models discussed here
along with the results obtained from previous studies indicate a good understanding for primary fossil sources in Houston. Because motor vehicles also emit precursors to SOA such as alkanes, light aromatics, and PAHs (Gentner et al., 2012), they likely contribute to anthropogenic SOA.





Biomass burning contributions to OC were small in comparison to fossil sources. CMB apportioned 5% of OC to BB. Because the CMB model utilized a BB profile collected near the source emissions, this value represents primary BB emissions. The selected source profile of open burning of pine forests (Lee et al., 2005) was considered to be the most representative of the available profiles because BB influences on Houston were traced back to open burning in the Yucatan Peninsula of Mexico (Fig. S7, S8). Importantly, the open burning profile has a relatively low levoglucosan-to-OC ratio compared to other BB profiles (Stone et al., 2009), making it an upper-estimate of BB contributions to OC. MM-PMF apportioned 11% of OC to BB; this value is expected to include both primary and secondary aerosol associated with BB (section 3.5.2.). Evidence of SOA from BB is indicated by the similarities in the MM-PMF BB factor profile BB and aged BB plumes (section 3.5.2). The difference between the estimates of primary and secondary BB organic aerosol (by MM-PMF) and primary BB organic aerosol (CMB) is used to estimate the magnitude of BB-derived SOA at an average of 6% of OC. AMS-PMF did not resolve a BB factor, perhaps due to the loss of AMS BB signatures ($m/z$ 60) during transport (Cubison et al., 2011) or due to the inclusion of BB in the CI-SV-OOA factor that had a low but significant correlation with CMB BB ($r=0.380$, $p=0.038$). The combination of CMB and MM-PMF provides separate estimates of primary and secondary BB contributions to OC, which cannot be resolved using either model alone.

Cooking was determined to have a minor, but uncertain contribution to $PM_{2.5}$. By MM-PMF, cooking was found to contribute to 1% of $PM_{2.5}$ OC, but with a large relative uncertainty (Fig. S10). The large relative error reflects the high degree of uncertainty in the estimation of this source contribution. Although AMS-PMF did not resolve a cooking factor, AMS-signatures of cooking and CI-SV-OOA correlated significantly, suggesting a cooking influence on this factor. The cooking contribution to CI-SV-OOA and OA, however, could not be resolved. Some degree of cooking influence on OA in Houston is expected due to the ubiquity of this source and a study near the HSC area, particularly for northerly winds (Wallace et al., 2018). The predominately southerly winds during this study (Fig S4) were associated with relatively small source contributions from cooking. Large variabilities across different studies are expected based on the sampling proximity to cooking sources and the prevailing wind direction. Cooking remains a difficult source to evaluate with receptor-based source apportionment models that require either fixed source profiles (in the case of CMB) or that resolve factors with consistent chemical composition (in the case of PMF). Cooking contributions are estimated by receptor-based models by way of a few molecular markers or AMS-signatures that may not represent the diversity of cooking activities that occur within an airshed. Consequently, these models cannot capture the inherent diurnal and spatial variability of cooking emissions. A better understanding of the variability of cooking emissions and model constraints are needed to lower uncertainties associated with contributions of cooking to ambient $PM_{2.5}$ organic aerosol.

Biogenic SOA was estimated to contribute up to 11% of organic aerosol. Among biogenic precursors to SOA, isoprene was determined to have the largest contribution. By MM-PMF isoprene SOA contributed 11% of $PM_{2.5}$ OC. This value is similar in magnitude to the 7% of $PM_{2.5}$ OC attributed to isoprene SOA by CMB that relied upon the isoprene tracer-to-OC ratios observed in chamber experiments (Kleindienst et al., 2007). The OC contribution estimated by PMF is expected to be more reliable for the Houston airshed, as it was derived from ambient measurements, rather than chamber experiments. Monoterpene SOA was found to be significantly lower, with a 0.5% contribution of α-pinene to OC resolved by CMB. Monoterpene SOA was resolved by neither MM-PMF nor AMS-PMF, likely due to an overall small contribution to OC or a lack of tracers to accurately define this source. Although AMS-PMF did not resolve an isoprene OA factor, moderate correlations between CI-SV-OOA and CMB estimates of isoprene SOA ($r=0.637$, $p<0.001$) and MM-PMF estimates of isoprene SOA ($r=0.626$, $p<0.001$) suggest that biogenic SOA was grouped with this factor. At small contributions to OC, CMB can better distinguish monoterpene contributions to OC than multivariate models. Overall, the SOA contribution from biogenic precursors are estimated in the range of 11% of OC but would be less if isoprene also has significant anthropogenic sources in this study domain (Fig. S9).

Anthropogenic SOA was identified as a major contributor to organic aerosol. By MM-PMF, two anthropogenic SOA factors were resolved: high-$NO_x$ anthropogenic SOA (6.6%) and low-$NO_x$ anthropogenic SOA (21.7%). MM-PMF





holds the advantage of identifying anthropogenic SOA sources based on unique tracers (Al-Naiema and Stone, 2017) but does not assume that tracer-to-OC ratios observed in chamber experiments are equivalent to the studied urban environment. Because it analyzes the co-variation of species over time, MM-PMF can capture anthropogenic SOA from other precursors that co-vary in time, even if they have a different precursor and are not defined by tracers in the source apportionment model (e.g., alkanes). As with biogenic SOA, the multivariate estimate of anthropogenic SOA is considered to be more reliable because it was developed from ambient measurements within the airshed and is expected to include SOA from a wide range of anthropogenic VOC. Using available profiles for anthropogenic SOA in CMB, 3% of OC was attributed to monoaromatic-derived SOA (e.g., benzene, toluene) and 5% was attributed to naphthalene-derived SOA. Larger SOA contributions from PAHs compared to monoaromatic precursors agrees with controlled chamber photo-oxidation of diesel exhaust, in which PAH-derived SOA was estimated to account for up to 54% to the total SOA mass formed in the first 12 h of oxidation (Chan et al., 2009). The total PAH contribution to SOA in HSC is expected to be larger, since other 2-3 ring PAHs with SOA-forming potential (Shakya and Griffin, 2010) are co-emitted with naphthalene; however, molecular tracers for larger PAH oxidation products are not yet defined, so the associated OC is unapportioned by the CMB model. The CMB-based estimate of anthropogenic SOA is limited to VOC precursors for which SOA profiles are available, to tracers that have been identified and are known to be specific to this source, and by the representativeness of the SOA profiles to the study domain. Thus, anthropogenic SOA from precursors other than monoaromatics and naphthalene are not apportioned by CMB. For these reasons, CMB is unable to provide a reliable estimation of *total* anthropogenic SOA, but CMB results yields specificity in relative source contributions for VOC that have been characterized in chamber experiments. The AMS-PMF LV-OOA factor that accounts for an average of 32% of $PM_1$ OC is expected to be influenced by anthropogenic SOA based on the correlation observed with high-$NO_x$ anthropogenic SOA in MM-PMF (r=0.515, p=0.004). AMS-PMF is unable to distinguish between anthropogenic and biogenic origins of SOA, in the absence of a strong signal that can identify a specific VOC precursor (Wallace et al., 2018; Xu et al., 2015). Relying on MM-PMF as the best estimate of anthropogenic SOA, we estimate that this source contributes an average of 28% of $PM_{2.5}$ OC, making it a major aerosol source in HSC second only to primary fossil fuel emissions. This novel combination and integration of *in situ* AMS-PMF, MM-PMF, and CMB modelsreveals the important contributions of anthropogenic SOA to $PM_{2.5}$ OC in the HSC area of Houston.

## 4 Conclusions

Comprehensive chemical analysis of fine PM in the HSC indicated a large contribution from sulfate and carbonaceous aerosol, as evidenced by offline filter-based measurements of $PM_{2.5}$ and *in situ* analysis of NR-$PM_1$. The novel combination of three source apportionment models (CMB, MM-PMF, and AMS-PMF) with statistical analyses provides a robust prediction of sources of OC, as well as the relative abundances of biogenic and anthropogenic SOA and the pathways by which they form. Together, these models were used to estimate primary sources of OC that included fossil sources (37-49%), from BB (5%), and cooking (1%).

Prior studies have recognized the large contribution from primary fossil sources to $PM_{2.5}$ in the HSC area but did not define sources of SOA. Here we show that secondary aerosols from anthropogenic origins contribute 28% of OC and are largely originated from precursors emitted from primary fossil sources. Anthropogenic SOA is among the largest sources of $PM_{2.5}$ OC near the HSC, while other SOA precursors—biogenic VOC (11%) and BB (6%)—have smaller contributions in comparison. Constraining the amount of SOA from BB and anthropogenic SOA is particularly significant because these two source categories have previously been difficult to estimate using one source apportionment method. To better define BB and anthropogenic SOA, future efforts should be placed on identifying and quantifying molecular markers to identify the specific precursors and pathways responsible for SOA formation. In this way, air quality management can target specific precursors and oxidants to reduce $PM_{2.5}$ levels. In the case of anthropogenic SOA, molecular tracers are needed for



recognized SOA precursors like n-alkanes, alcohols, and PAHs (beyond naphthalene). Well-defined SOA profiles for these precursors will support CMB-based methods and aid in the interpretation of MM-PMF results.

The analytical approaches and source apportionment methods presented herein can be applied elsewhere to develop a better understanding of BB and anthropogenic VOC to SOA. Combining multiple source apportionment techniques

overcomes limitations of using these receptor models in isolation. For instance, BB SOA contributions to organic aerosol by can be estimated by subtracting primary BB estimated by CMB from the sum of primary and secondary BB from MM-PMF. This method is expected to be accurate when the chemical nature of the primary biomass emissions is known and a representative chemical profile is used. This approach can overcome previous limitations on constraining BB-derived SOA, which was challenged by the large number and variability of its precursors and the lack of knowledge of its major SOA

products. In addition, the use of aromatic SOA tracers constrains MM-PMF  estimations of anthropogenic SOA and overcomes limitations in source profiles for use in CMB modeling. In order to apportion anthropogenic SOA, it is necessary to explicitly include anthropogenic SOA tracers as fitting species in MM-PMF. While initial guidance on anthropogenic SOA tracer selection is provided elsewhere (Al-Naiema and Stone, 2017), the utilized tracers should be expanded as anthrhopogenic SOA becomes more chemically-defined. As more anthropogenic SOA tracers are identified and incorporated

in to MM-PMF modeling, greater specificity as to the major precursor gases may be gained. The methodological approach presented here can be used to gain insight to sources of $PM_{2.5}$ in diverse urban environments that can inform strategies to manage urban air quality, particularly in areas that exceed air quality standards or guidelines.

**Acknowledgements**

This research was supported by the National Science Foundation (NSF) through AGS grant number 1405014. We thank the Texas Commission on Environmental Quality and City of Houston for access to the Clinton Drive monitoring site.  The Rice University group acknowledges funding from the Houston Endowment, the assistance of collaborators from the University of Houston, and Loredana Suciu for assistance in changing filters.



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





**Figure Captions**

**Figure 1**: A summary of NR-PM$_1$ measurements at Clinton Drive for the period May 13$^{th}$ – 29$^{th}$, 2015; a) Time series including organics, sulfate, ammonium, nitrate and chloride, b) average percent contributions, and c) the hourly diurnal

profile (daytime hours are highlighted in yellow). The bottom whisker, bottom box line, top box line and top whisker indicate the 5$^{th}$, 25$^{th}$, 75$^{th}$ and 95$^{th}$ percentiles, respectively. Lines inside the boxes represent the hourly median of the data. One high data point of organic (57.42 µg m$^{-3}$) on 23/5/2015 at 14:05 LT is not shown.

**Figure 2**: Day and night PM$_{2.5}$ mass composition at Clinton Drive for the period 5-27 May 2015. Carbonaceous and ionic

species account for 82% of PM$_{2.5}$, on average, and the remaining 18% was attributed to other species that were not measured, such as silica, alumina and metals. PM$_{2.5}$ mass (average ± analytical uncertainty) was obtained using TEOM from TCEQ.

**Figure 3**: Scatter plot of the molar concentration of sulfate plus nitrate versus ammonium, measured by HR-ToF-AMS during May 2015. The higher anions relative to ammonium suggest that ammonium molar concentrations are insufficient to

fully neutralize sulfate, indicating acidic aerosols.

**Figure 4**: Mass spectra of PMF factors in PM$_1$ OA at Clinton Drive during May 2015.

**Figure 5**: AMS-PMF factors identified in PM1 OA; a) time series of OA factors (HOA, CI-SV-OOA, and LV-OOA) and b)

diurnal profiles of OA factor mass concentrations. Bottom whisker, bottom box line, top box line and top whisker indicate the 5th, 25th, 75th and 95th percentiles, respectively. Lines inside the boxes represent the hourly median and circles represent the hourly mean.

**Figure 6**: Source contributions to PM$_{2.5}$ OC at Clinton Drive during May 2015 estimated by CMB modeling.

**Figure 7**: MM-PMF factor profiles. The y-axis represents the percentage of species attributed to each factor.

**Figure 8**: MM-PMF factor contributions to PM$_{2.5}$ OC at Clinton Drive in May 2015. Samples labeled D were collected during daytime (7:00 - 18:00 LT) and those labeled N samples were collected during nighttime (19:00 - 6:00 LT).

**Figure 9**: Summary of the average source contributions to PM OC at Clinton Drive determined for (a) NR-PM$_1$ by AMS-PMF, and (b) PM$_{2.5}$ by MM-PMF and CMB, including primary fossil sources (green), biomass burning (BB; yellow), cooking (orange), biogenic secondary organic carbon (BSOA; purple) and anthropogenic secondary organic carbon (ASOA; dark grey). Numerical values presented in this figure are summarized in Table 5.

40





Figure 1

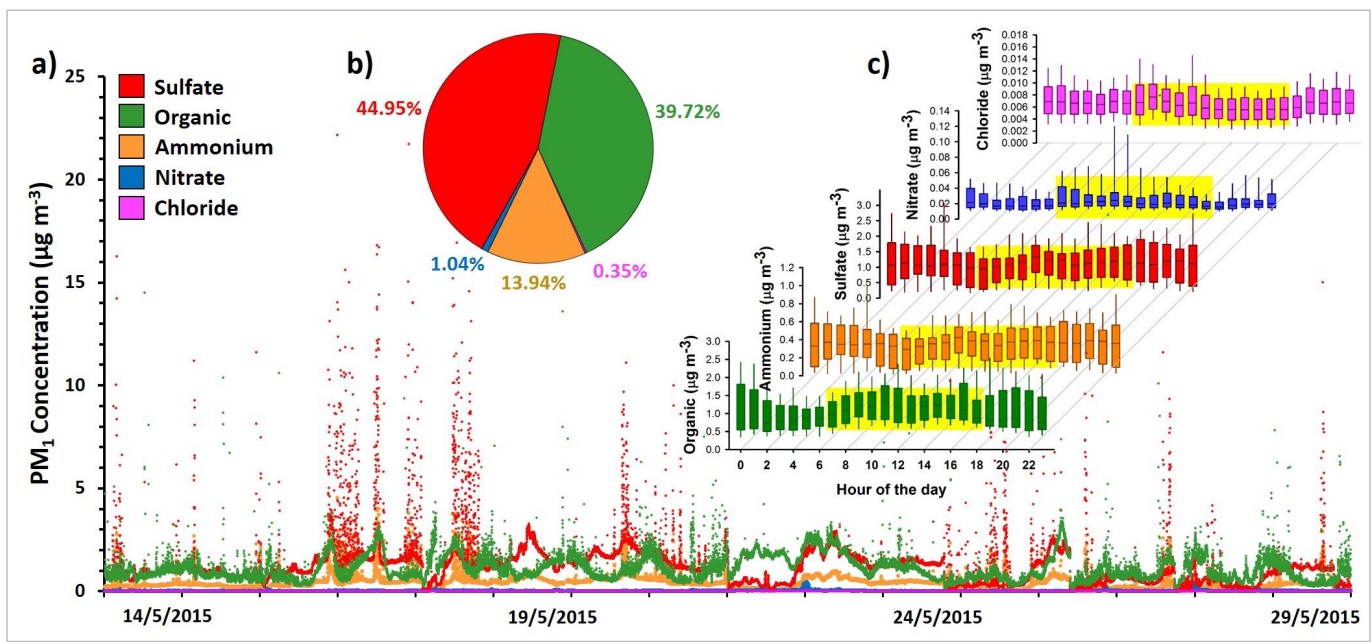



**Figure 2**

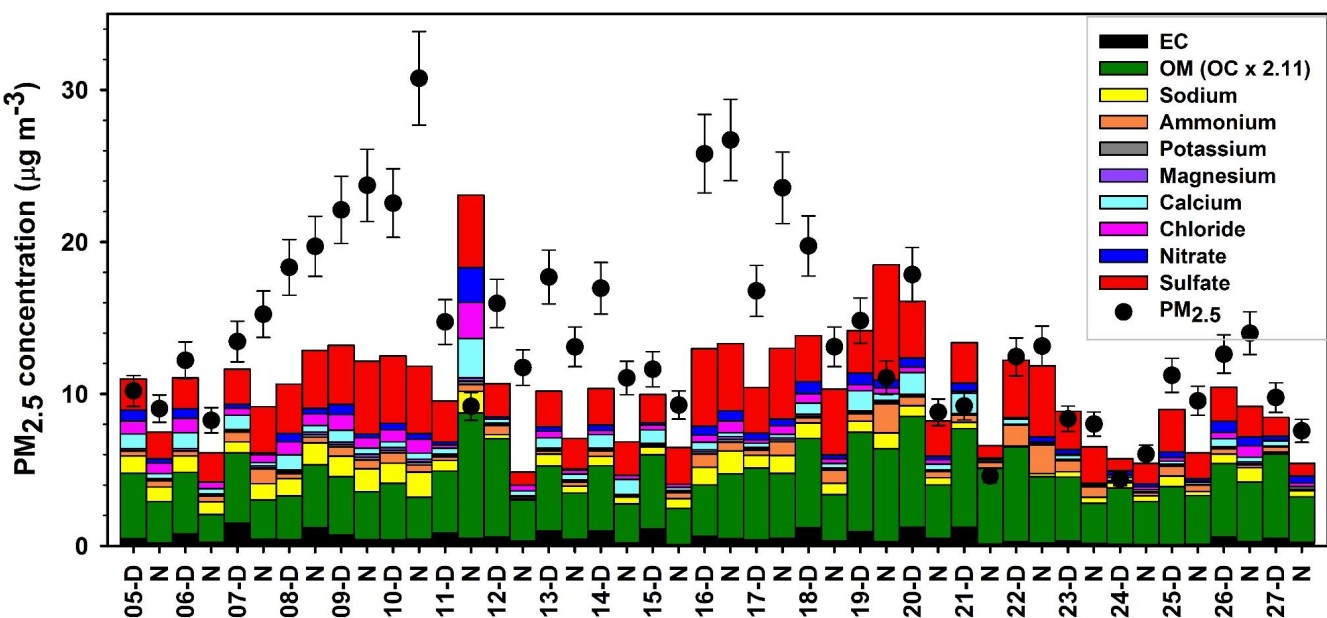



**Figure 3**:

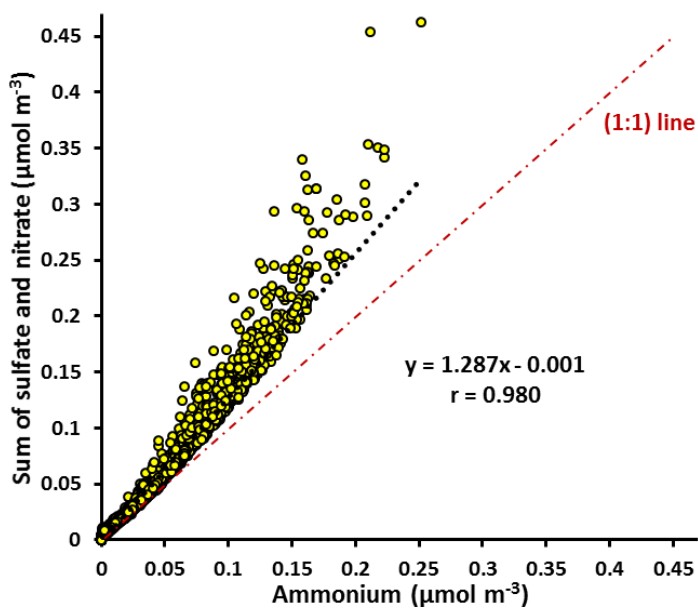





**Figure 4**:

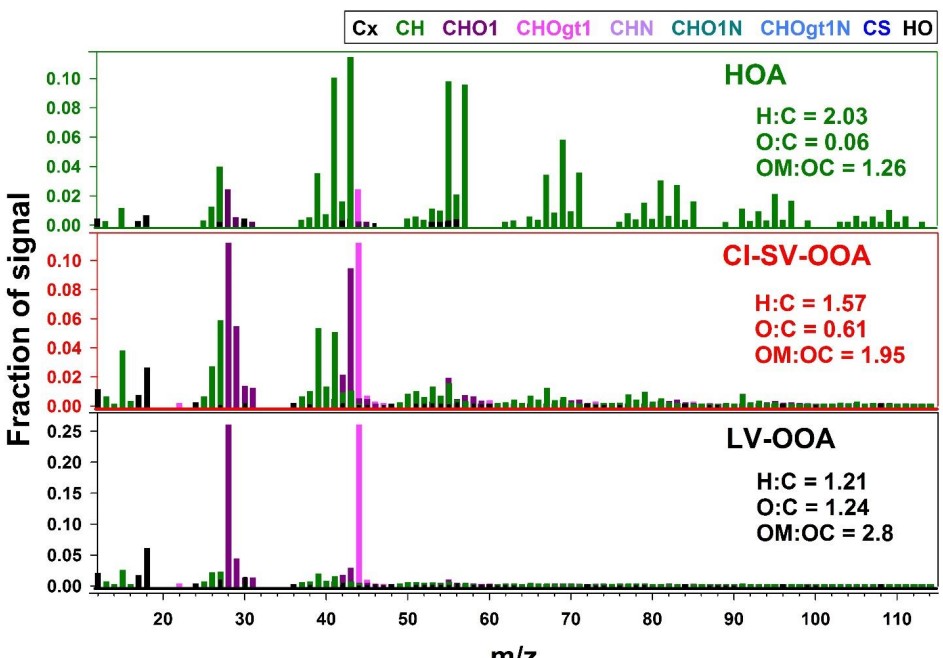





**Figure 5**:

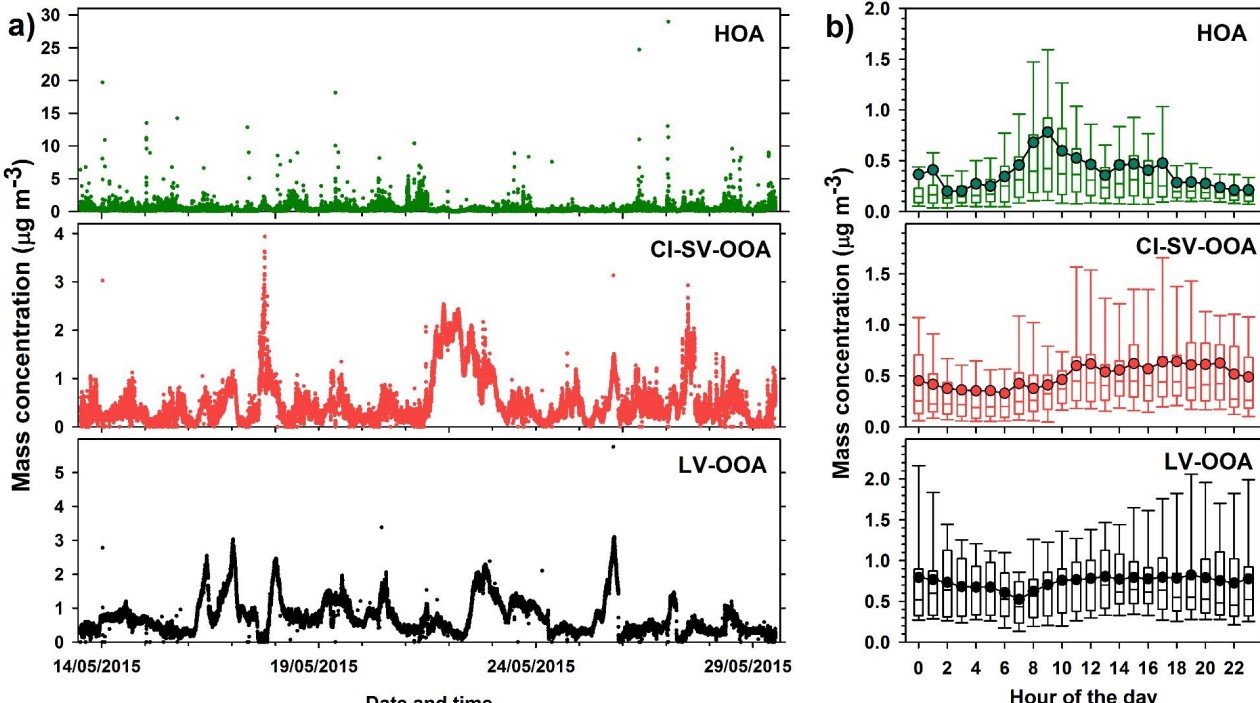





**Figure 6**:

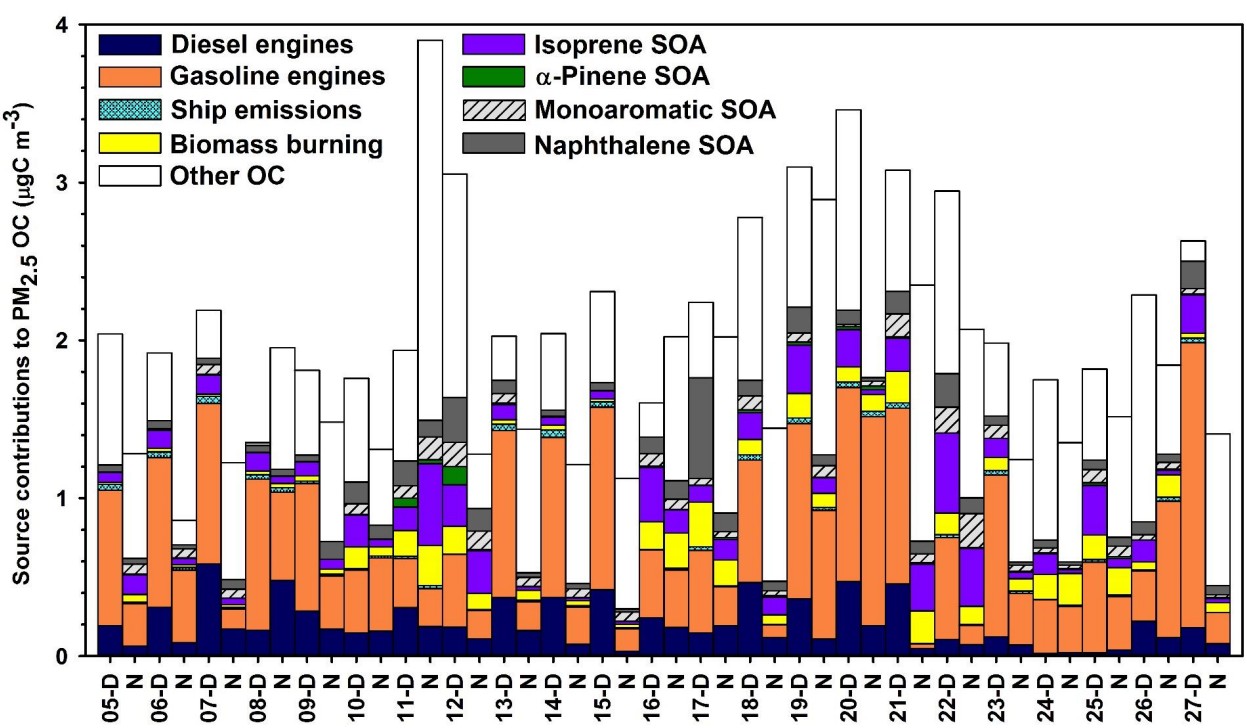





**Figure 7**:

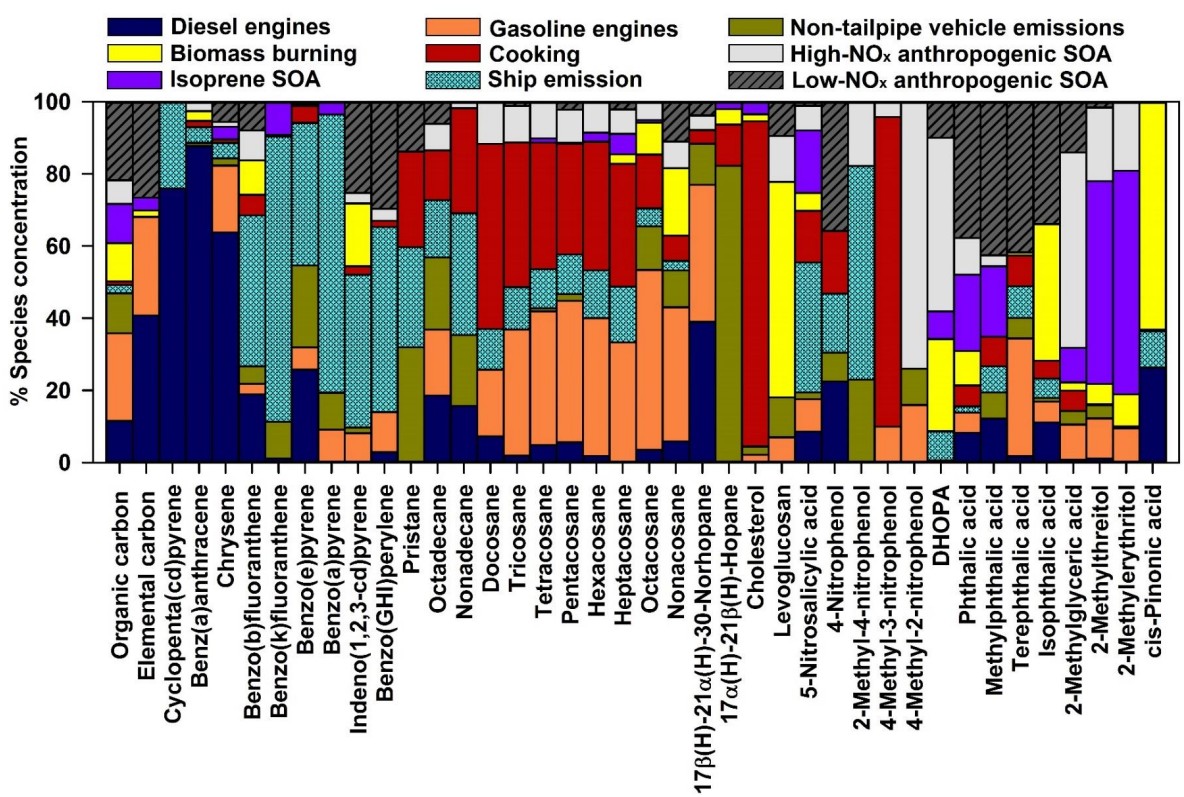





**Figure 8**:

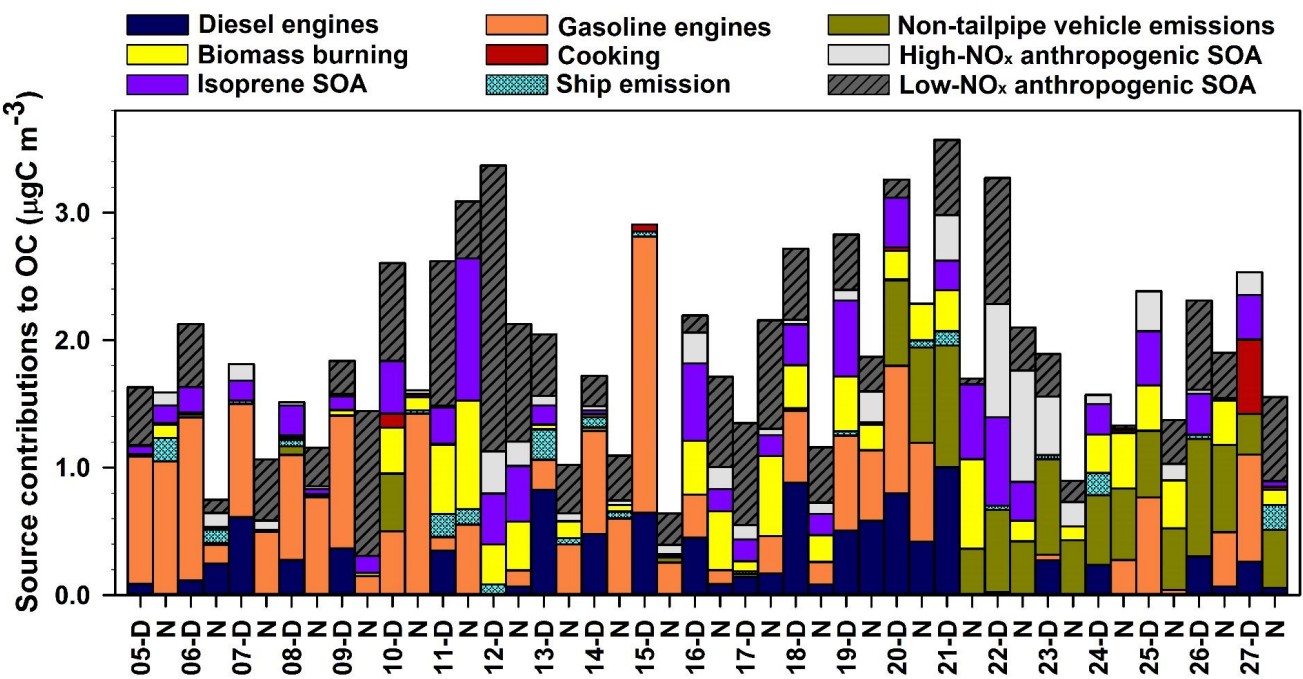




**Figure 9**:

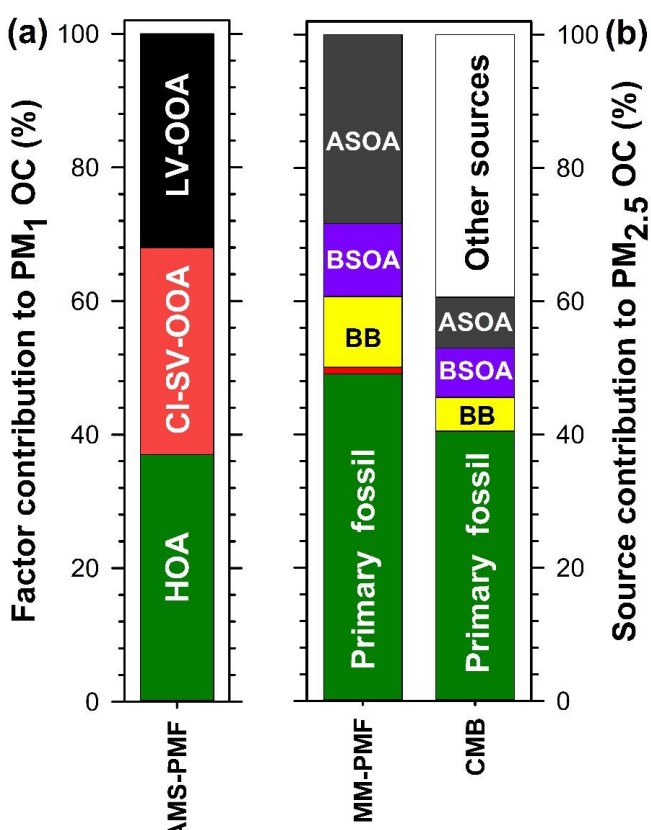





**Table 1.** Summary of ToF-AMS high-resolution (1 min) results of NR-PM$_1$ composition ($\mu$g m$^{-3}$), measured in HSC for the period 13-29 May, 2015.

|  | Mean | Median | Stdev. | Max | Min |
|---|---|---|---|---|---|
| Organic | 1.14 | 0.95 | 0.8 | 57.42 | 0.06 |
| Nitrate | 0.03 | 0.02 | 0.03 | 0.49 | BDL |
| Ammonium | 0.4 | 0.36 | 0.35 | 4.54 | BDL |
| Sulfate | 1.29 | 1.12 | 1.23 | 22.16 | BDL |
| Chloride | 0.01 | 0.01 | 0 | 0.05 | BDL |





**Table 2.** Summary of the elemental analysis of NR-PM$_1$organic aerosol. $\overline{OSc}$ is defined as the average oxidation state (Kroll et al., 2011) and is calculated by 2*O:C - H:C.

| Elemental Ratio | Mean | Median | Stdev. |
|---|---|---|---|
| OM:OC | 2.11 | 2.12 | 0.29 |
| O:C | 0.72 | 0.73 | 0.22 |
| H:C | 1.5 | 1.52 | 0.17 |
| $\overline{OSc}$ | -0.09 | -0.07 | 0.6 |



5  **Table 3**: PM$_{2..5}$ its major components (µg m$^{-3}$) and estimated pH for daytime, nighttime, and overall periods (± standard deviation) at Clinton Drive during May 2015. Organic matter (OM) was estimated based on the mean OM:OC ratio of 2.11 obtained from HR-ToF-AMS (Table 2). P-values ≤ 0.05 indicate that the difference between daytime and nighttime concentrations are statistically significant at the 95% confidence interval.

| | Daytime | Nighttime | Overall | P-value |
|---|---|---|---|---|
| PM$_{2.5}$ | 14.73 ± 5.08 | 13.35 ± 6.93 | 14.04 ± 6.05 | 0.447 |
| Elemental carbon (EC) | 0.71 ± 0.38 | 0.34 ± 0.24 | 0.52 ± 0.37 | < 0.001 |
| Organic matter | 4.78 ± 1.19 | 3.57 ± 1.40 | 4.17 ± 1.42 | 0.011 |
| Sodium | 0.73 ± 0.37 | 0.79 ± 0.50 | 0.76 ± 0.43 | 0.566 |
| Ammonium | 0.47 ± 0.28 | 0.56 ± 0.49 | 0.52 ± 0.40 | 0.489 |
| Potassium | 0.08 ± 0.04 | 0.08 ± 0.06 | 0.08 ± 0.05 | 0.703 |
| Magnesium | 0.10 ± 0.04 | 0.10 ± 0.06 | 0.10 ± 0.05 | 0.838 |
| Calcium | 0.68 ± 0.39 | 0.40 ± 0.52 | 0.54 ± 0.47 | 0.044 |
| Chloride | 0.40 ± 0.31 | 0.48 ± 0.49 | 0.44 ± 0.40 | 0.512 |
| Nitrate | 0.45 ± 0.22 | 0.34 ± 0.47 | 0.40 ± 0.37 | 0.353 |
| Sulfate | 2.74 ± 0.98 | 3.00 ± 1.71 | 2.87 ± 1.39 | 0.529 |
| Aerosol pH | 0.54 ± 1.45 | 0.32 ± 0.81 | 0.44 ± 0.39 | 0.075 |





**Table 4**: Chemical mass balance (CMB) estimates of the absolute relative source contributions to $PM_{2.5}$ OC at Clinton Drive in May 2015 averaged over daytime, nighttime, and all periods. P-values $\leq 0.05$ indicate that the difference between daytime and nighttime source contributions are statistically significant at the 95% confidence interval.

| Source category | Daytime | | Nighttime | | Overall | | p-value |
|---|---|---|---|---|---|---|---|
| | ($\mu$gC m$^{-3}$) | (% OC) | ($\mu$gC m$^{-3}$) | (% OC) | ($\mu$gC m$^{-3}$) | (% OC) | |
| Diesel engines | 0.27 ± 0.15 | 11.82 ± 6.32 | 0.13 ± 0.09 | 7.80 ± 5.14 | 0.20 ± 0.14 | 9.81 ± 6.05 | 0.001 |
| Gasoline engine [1] | 0.82 ± 0.37 | 37.14 ± 16.29 | 0.36 ± 0.30 | 22.56 ± 17.77 | 0.59 ± 0.41 | 29.80 ± 18.38 | < 0.001 |
| Ship emission | 0.02 ± 0.01 | 1.11 ± 0.68 | 0.01 ± 0.01 | 0.73 ± 0.57 | 0.02 ± 0.01 | 0.92 ± 0.65 | 0.001 |
| Biomass burning | 0.10 ± 0.08 | 4.49 ± 3.60 | 0.10 ± 0.07 | 5.70 ± 3.52 | 0.10 ± 0.07 | 5.09 ± 3.57 | 0.627 |
| Isoprene SOA | 0.18 ± 0.11 | 7.99 ± 4.86 | 0.11 ± 0.13 | 5.90 ± 5.48 | 0.15 ± 0.12 | 6.95 ± 5.23 | 0.063 |
| α-Pinene SOA | 0.01 ± 0.03 | 0.59 ± 0.92 | 0.01 ± 0.01 | 0.38 ± 0.33 | 0.01 ± 0.02 | 0.49 ± 0.69 | 0.172 |
| Monoaromatic SOA | 0.06 ± 0.05 | 2.37 ± 1.95 | 0.06 ± 0.05 | 3.46 ± 2.63 | 0.06 ± 0.05 | 2.91 ± 2.35 | 0.966 |
| Naphthalene SOA | 0.12 ± 0.13 | 5.36 ± 5.50 | 0.06 ± 0.04 | 3.91 ± 2.36 | 0.09 ± 0.10 | 4.63 ± 4.25 | 0.044 |
| Other OC | 0.68 ± 0.37 | 29.24 ± 15.13 | 0.36 ± 0.30 | 49.56± 18.58 | 0.59 ± 0.41 | 39.40 ± 19.65 | 0.235 |

[1] Gasoline engines factor represent the sum of the contribution from smoking and non-smoking gasoline engines



**Table 5**: Summary of the average source contributions to OC (%) for each of the three source apportionment models. Missing values correspond to sources that were either not included in the model (i.e. CMB) or not resolved by the model (i.e. PMF). Contributions of $PM_1$ factors to OC were estimated from the OA contributions and OM:OC ratios for each factor (Fig. 4).

| Source category / factor | CMB (PM$_{2.5}$) | MM-PMF (PM$_{2.5}$) | AMS-PMF (PM$_1$) |
|---|---|---|---|
| **Primary fossil** | 41[a] | 49[b] | |
| **Cooking** | | 1 | |
| **Biomass burning (BB)** | 5 | 11 | |
| **Biogenic SOA (BSOA)** | 7 | 11 | |
| **Anthropogenic SOA (ASOA)** | 8 | 28 | |
| **Other OC** | 39[c] | | |
| **HOA** | | | 37 |
| **CI-SV-OOA** | | | 31 |
| **LV-OOA** | | | 32 |

[a] Primary fossil sources from the CMB model were calculated as the sum of diesel engines, gasoline engines, and ship emissions; [b] Primary fossil sources of MM-PMF were calculated as the sum of diesel engines, gasoline engines, non-tailpipe emissions, and ship emissions; [c] Other OC sources in CMB represent the fraction of OC that was not apportioned by CMB model. In the absence of unidentified primary sources in CMB, other OC represents SOA.