# Peer review of "Source apportionment of fine particulate matter in Houston, Texas: Insights to secondary organic aerosols"

_Atmospheric Chemistry and Physics, 2018_

## Short Comment (SC1) · 16 Apr 2018

Dear Ibrahim, Elizabeth and co-authors,

This is a very great work providing a comparison of OA source apportionment results obtained using PMF-AMS, PMF-filter based (MM-PMF) and CMB. There are only few papers in the literature showing such direct comparison and it is interesting to show how they agree.

I would like to suggest you to have a look to 2 papers that we have very recently published about the use of primary and secondary organic molecular markers in PMF

source apportionment. This is maybe helpful.

Srivastava, D., Tomaz, S., Favez, O., Lanzafame, G. M., Golly, B., Besombes, J.-L., Alleman, L. Y., Jaffrezo, J.-L., Jacob, V., Perraudin, E., Villenave, E. and Albinet, A.: Speciation of organic fraction does matter for source apportionment. Part 1: A one-year campaign in Grenoble (France), Science of The Total Environment, 624, 1598–1611, doi:10.1016/j.scitotenv.2017.12.135, 2018a.

Srivastava, D., Favez, O., Bonnaire, N., Lucarelli, F., Haeffelin, M., Perraudin, E., Gros, V., Villenave, E. and Albinet, A.: Speciation of organic fractions does matter for aerosol source apportionment. Part 2: Intensive short-term campaign in the Paris area (France), Science of The Total Environment, 634, 267–278, doi:10.1016/j.scitotenv.2018.03.296, 2018b.

Regards,

Alexandre

---

## Referee Comment (RC1) · Anonymous Referee #1 · 17 Jul 2018

This paper presents source apportionment results from a comprehensive on-line and off-line chemical datasets collected concurrently in Houston, Texas. The authors applied three different for source apportionment approaches to determine the sources and their contributions, which has not be done before to my knowledge. The source apportionment results were compared between the three approaches and their finding that the primary source contributions agreed was encouraging. Furthermore, it allowed for more in-depth characterization of the different sources of SOA by combining the results from the three methods, which will be of interest to many. My main comment would be that perhaps the authors could recommend a tracer for biogenic and anthropogenic SOA. I may have missed it but their comprehensive dataset might allow at

least for a tentative proposal as applying all three source apportionment methods will not be feasible in many cases and may allow the results from this study to be applied more widely. The paper is well written and logically set out and in my opinion fits within the scope of ACP.

I have a few minor comments below that the authors may wish to consider.

1. Section 3.2: did you do a PM2.5 mass balance, comparing the measured PM2.5 (gravimetric) against the reconstructed PM2.5 mass concentration from the chemical analysis?

2. Page 9, line 37: Can you say there is a cooking influence in CI-SV-OOA if there is evening peak in the diurnal profile? Normally, a peak associated with evening meal times is a marker for cooking emissions. Without I am not sure that there is much influence from cooking, especially as your MM-PMF analysis only apportions 1% of the PM2.5 to cooking. Perhaps this is more of SV-OOA factor with some hydrocarbon/primary local emission influence.

3. Page 10, line 19: In your CMB results you have said that the unclassified OC is likely SOA, would you expect more SOA at night (49%) compared to daytime (29%)? As you have already apportioned Biogenic and anthropogenic SOA in the model, I am guessing that this SOA is regional in nature, and so I would not expect such a difference day/night.

---

## Referee Comment (RC2) · Anonymous Referee #2 · 24 Jul 2018

**General comments**

This manuscript reports a compositional analysis of $PM_{2.5}$ by offline filter measurement on a day-night basis and $NR\text{-}PM_1$ by an in-situ HR-TOF-AMS near the HSC in May 2015. Integrated source apportionment analysis for OC was conducted incorporating AMS-PMF, MM-PMF and CMB methods. The three source apportionment models are in agreement that ~50% of OC was formed from primary fossil fuel combustion. SOA was further apportioned to anthropogenic and biogenic sources by combing results from MM-PMF and CMB. The comprehensive nature of this work in data and source apportionment analysis bring to the literature a valuable case study report. This work nicely contributes to our improved understanding of SOA sources. I have the following specific points that need clarification/more discussion in their next revision.

**Major comments:**

1. The offline and online measurements overlapped between 13-27 May. Some comparison of the two measurements would be useful as data cross checking and verification. Such comparisons are currently lacking. For example, what's the percentage of $PM_1$ contributed to total $PM_{2.5}$ and how the two mass concentrations correlated with each other? This is important not only for data validation, but also for the latter comparison between AMS-PMF and MM-PMF and CMB.

2. There are inconsistencies among the source apportionment results by the different approaches. AMS-PMF resolved a cooking influenced factor but no BB related OA, and the authors proposed the loss of the m/z 60 signatures during transport as possible reason. This reason is not supported by their data, as MM-PMF resolved both BB and cooking sources and BB even contributed more than cooking emission (11% vs 1%). It is odd that the authors didn't include cooking emission profiles in CMB. Previous studies have successfully apportioned $PM_{2.5}$ to cooking emission (6%) in *Fraser e al.* [2003], as mentioned in the introduction part.

3. The AMS-PMF resolved a CI-SV-OOA factor that contributed to 31% of ambient OC (Table 5), but such a considerable contribution seems inconsistent with the minor cooking contribution estimated from MM-PMF (1%). This inconsistency calls into question whether naming the SV-OOA factor as "cooking-influenced" is appropriate, as it may imply this factor is largely influenced by cooking emissions.

4. The identification of different vehicle emission factors, i.e. diesel engines, gasoline engines and non-tailpipe vehicle emissions, is not very convincing, considering alkanes, PAHs are not unique source tracers to vehicle emissions. It is unusual that norhopane and hopane are separated into two different factors. The two species are usually highly correlated with each other. What's the correlation between the two species in this study? Also, a high amount of EC was present in low-NOx anthropogenic SOA factor. Does this indicate the mixing of primary sources in this factor?

5. In the CMB model (section 2.5), were EC and levoglucosan included in the calculation? If not, which species was/were mainly responsible for determining the contributions from diesel engines and biomass burning? Also, it would be better if the statistical performance of the CMB results can be reported (e.g., $\chi 2$, calculated vs. modelled species concentrations, species source contributions).

6. The sample size of MM-PMF for PM2.5 OC is 46 in the study, but in the introduction section it is suggested the sample size should be 60–200 (P.2 line 39–40). It seems, because of this reason, the MM-PMF performance is not very robust as shown in Table S2 (some key species like cis-pinonic acid and phthalic acid are not well modelled) and Table S5 (only 54% of bootstrap BB factor was mapped). The uncertain BB factor contribution in MM-PMF would also weaken the reliability of BB SOA estimation in this study. Please comment on how the small sample size affect the MM-PMF results in this work.

7. Fig. 8 shows that the non-tailpipe vehicle emissions (marked largely by 17α(H),21β(H)-hopane only) made a notable contribution after 20 May (except 10 May daytime), is there any reason for it?

8. Is there any reason to select 17β(H)-21α(H)-30-norhopane over 17α(H),21β(H)-30-norhopane as the vehicle emissions tracer in CMB and MM-PMF? The former species should be less abundant in ambient and source PM, and therefore is less commonly used in receptor models (e.g., Yu et al., 2011, Analytical and Bioanalytical Chemistry, 401, 3125-3139).

9. When the source apportionment results from AMS-PMF, MM-PMF and CMB were integrated to obtain insights in SOA contributions (section 3.5), it seems the conclusion that anthropogenic SOA is the dominant contributor is largely drawn from the MM-PMF results, while the CMB and AMS-PMF results do not converge to the same conclusion (e.g., P.14 line 17: CMB is unable to provide a reliable estimation of total anthropogenic SOA, … ; and line 20–21: AMS-PMF is unable to distinguish between anthropogenic and biogenic origins of SOA, …). However, the MM-PMF results may have large uncertainties, especially considering the small sample size and poor stability (e.g., only 67% of bootstrap high-NOx anthropogenic SOA factor can be mapped as shown in Table S5), making such a conclusion less convincing. Additional supportive evidence/argument for this point is recommended.

**Minor comments:**

1. Line 4-5 in p7: "organic carbon", "elemental carbon" and "Organic matter" should be removed as the abbreviations have been introduced before. Check throughout the context to avoid redundant words.
2. Include the measured OC concentrations in Figure 8, which can help visualize directly how MM-PMF predicts the measured concentrations.
3. Line 16 in p1: Change "fine" to "ambient".
4. Line 25 in p1 and line 11 in p2: "VOC" should be "VOCs".
5. Line 21 in p2: "for" should be "from".
6. Line 13-16 in p6: the correlations are R or $R^2$?
7. Figure S10, correct the typo mistake for the *y*-axis.
8. P.5 line 36: Bituminous coal source profile was said to be included in CMB, but it is not reported in the result section (3.4.1), please check.
9. P.8 line 25–26: The order of HOA, CI-SV-OOA and LV-OOA should be reversed to be consistent with the statement in the next sentence, as well as the abstract (line 20–22).
10. p.10 line 13: Full stop missing in "… comparing this study to Buzcu et al., (2006)."
11. P.11 line 34–36: Yan et al. (2008) did not report isophthalic acid in aged BB plumes, please consider citing another reference.
12. P.13 line 8 & 9: section 3.5.2 → section 3.4.2
13. P.13 line 33–34 argued that isoprene-derived SOA contribution estimated by MM-PMF is more reliable than that by CMB, is there any further support/reference for this argument? Uncertainties associated with ambient measurements and temporal correlation between species in PMF should not be neglected.
14. Space missing: after Table S2 in P. 10 line 28; between "models reveals" in P.14 line 25; after NR-PM1 in P.33 Table 2 title.
15. P.14 line 32: …, from BB (5%), …
16. P.15 line 5: … to organic aerosol by can be estimated by…
17. Missing close parenthesis for graph (a) and (b) in Fig. S3.
18. Misaligned unit in the y-axis title of Fig. S10.

---

## Author Comment (AC1) · 31 Aug 2018

Dear Alexandre,

Thank you for your feedback on this manuscript. We greatly appreciate your interest and suggestions to improve it. We agree that these papers are relevant to the work presented herein and have added reference to them in three places:

In the introduction, when describing the value of molecular-marker driven PMF for apportionment of SOA: "This approach has been particularly useful in the elucidation of SOA contributions to ambient PM, by providing insight into the precursors and path-

ways by which they form (Hettiyadura et al., 2018; Srivastava et al., 2018; Wang et al., 2017)."

In our recommendations for expanding tracers for PAH-derived SOA: "In prior MM-PMF studies in France, oxy-PAH and nitro-PAH have been useful in tracing SOA derived from larger PAH (Srivastava et al., 2018a, 2018b). "

In our recommendations for better constraining BB SOA: "Phenolic oxidation associated with BB SOA has also been identified using methyl-nitrocatechols (Srivastava et al., 2018a, 2018b)."

Thank you, Betsy

Works Cited

Hettiyadura, A. P. S., L. Xu, T. Jayarathne, K. Skog, H. Guo, R. J. Weber, A. Nenes, F. N. Keutsch, N. L. Ng and E. A. Stone, Source apportionment of organic carbon in Centreville, AL using organosulfates in organic tracer-based positive matrix factorization. Atmospheric Environment, 186, 74-88, doi:10.1016/j.atmosenv.2018.05.007, 2018.

Srivastava, D., Tomaz, S., Favez, O., Lanzafame, G. M., Golly, B., Besombes, J.-L., Alleman, L. Y., Jaffrezo, J.-L., Jacob, V., Perraudin, E., Villenave, E. and Albinet, A.: Speciation of organic fraction does matter for source apportionment. Part 1: A one-year campaign in Grenoble (France), Science of The Total Environment, 624, 1598–1611, doi:10.1016/j.scitotenv.2017.12.135, 2018a.

Srivastava, D., Favez, O., Bonnaire, N., Lucarelli, F., Haeffelin, M., Perraudin, E., Gros, V., Villenave, E. and Albinet, A.: Speciation of organic fractions does matter for aerosol source apportionment. Part 2: Intensive short-term campaign in the Paris area (France), Science of The Total Environment, 634, 267–278, doi:10.1016/j.scitotenv.2018.03.296, 2018b.

Wang, Q. Q., X. He, X. H. H. Huang, S. M. Griffith, Y. M. Feng, T. Zhang, Q. Y. Zhang, D. Wu and J. Z. Yu, Impact of Secondary Organic Aerosol Tracers on

Tracer-Based Source Apportionment of Organic Carbon and PM2.5: A Case Study in the Pearl River Delta, China. Acs Earth and Space Chemistry 1 (9), 562-571, doi:10.1021/acsearthspacechem.7b00088, 2017.
* * *

---

## Author Comment (AC2) · 1 Sep 2018

Referee #1 general comments: This paper presents source apportionment results from a comprehensive on-line and off-line chemical datasets collected concurrently in Houston, Texas. The authors applied three different for source apportionment approaches to determine the sources and their contributions, which has not be done before to my knowledge. The source apportionment results were compared between the three approaches and their finding that the primary source contributions agreed was encouraging. Furthermore, it allowed for more in-depth characterization of the different sources of SOA by combining the results from the three methods, which will be of inter-

est to many. My main comment would be that perhaps the authors could recommend a tracer for biogenic and anthropogenic SOA. I may have missed it but their comprehensive dataset might allow at least for a tentative proposal as applying all three source apportionment methods will not be feasible in many cases and may allow the results from this study to be applied more widely. The paper is well written and logically set out and in my opinion fits within the scope of ACP.

I have a few minor comments below that the authors may wish to consider.

Response to Referee #1 general comments: We thank the reviewer for their review of the manuscript. We agree with the reviewer's summary of this work. In regards to their main comment about providing recommendations for biogenic and anthropogenic SOA tracers for future source apportionment studies, we have significantly revised the conclusion section to provide recommendations for future studies. The revised text appears in the second-to-last paragraph in section 4 (lines 7-29, page 16):

"MM-PMF is a useful approach for estimating source contributions to OC and PM2.5, particularly when source profiles for sources are not available or are not well defined, which is often the case for SOA. In order to apportion anthropogenic SOA, it is necessary to explicitly include anthropogenic SOA tracers as fitting species in the PMF model. Initial guidance on anthropogenic SOA tracer selection was drawn from Al-Naiema and Stone (Al-Naiema and Stone, 2017). In this study, to track anthropogenic SOA formed from aromatic VOC under high NOx conditions, 4-methyl-2-nitrophenol and DHOPA served as key tracers. For PAH-derived SOA, key tracers were 4-nitrophenol, phthalic acid for naphthalene-derived SOA, and 4-methylphtalic acid for methylnaphthalene SOA. In prior MM-PMF studies in France, oxy-PAH and nitro-PAH have been useful in tracing SOA derived from larger PAH (Srivastava et al., 2018a; Srivastava et al., 2018b). The utilized tracers should be expanded as anthropogenic SOA becomes more chemically-defined. In particular, molecular tracers are needed for recognized SOA precursors that include other aromatic compounds, n-alkanes, alcohols, and PAHs (beyond naphthalene and its derivatives). While few biogenic SOA tracers

were detected in HSC, 2-methylerythritol and 2-methylthreitol were valuable in identifying the isoprene SOA factor. Caution should be used in the use of 2-methylglyceric acid that is a high-NOx SOA product formed from MACR that can come from biogenic or anthropogenic origins; while plants are the major source of isoprene globally, motor vehicles contribute the majority of the MACR in urban Houston (Park et al., 2011). Similarly, SOA from BB was identified by way of isopthalic acid and cis-pinonic acid, consistent with aged BB emissions documented in the literature (Yan et al., 2008); however, these compounds can also have other sources, such as primary emissions and monoterpene-derived SOA, respectively. Phenolic oxidation associated with BB SOA has also been identified using methyl-nitrocatechols (Srivastava et al., 2018a, 2018b). To better define BB and anthropogenic SOA, future efforts should be placed on identifying and quantifying molecular markers to identify the specific precursors and pathways responsible for SOA formation. Better definition of the molecular profiles of anthropogenic and BB SOA will support CMB-based methods and aid in the interpretation of MM-PMF results."

Referee #1 specific comment 1: Section 3.2: did you do a PM2.5 mass balance, comparing the measured PM2.5 (gravimetric) against the reconstructed PM2.5 mass concentration from the chemical analysis?

Response to Referee #1 specific comment 1: Yes, a mass balance was performed in which the PM2.5 mass measured by the TEOM at Clinton Drive was compared to the sum of the species measured on the filters, including organic carbon converted to organic matter, elemental carbon, and inorganic ions. These data are shown in Figure 2 and we have added a statement regarding these results in section 3.2 (lines 19-21, page 7): "On average, OM, EC, and inorganic ions accounted for 80% of the PMň2.5 mass (Fig. 2), with the remaining mass expected to arise from unmeasured species such as crustal metal oxides (e.g. silica, alumina), other metals, and particle-bound water."

Referee #1 specific comment 2: Page 9, line 37: Can you say there is a cooking

influence in CI-SV-OOA if there is evening peak in the diurnal profile? Normally, a peak associated with evening meal times is a marker for cooking emissions. Without I am not sure that there is much influence from cooking, especially as your MM-PMF analysis only apportions 1% of the PM2.5 to cooking. Perhaps this is more of SV-OOA factor with some hydrocarbon/primary local emission influence.

Response to Referee #1 specific comment 2: We agree with the reviewer that classifying this factor as SV-OOA would reflect its main nature; however, we also consider that the denomination of this factor should include a reference to the observed influence of cooking activities. This influence is evidenced by (i) statistically significant association with mass fragments reported as tracers of food cooking (Table R1), (ii) co-variability between CI-SV-OOA and the C3H3O+ mass fragment, typically used to distinguish cooking organic aerosol (COA) from HOA (Figure R1) (Mohr et al., 2012; Sun et al., 2016; Wallace et al., 2018) and (iii) m/z 55 to m/z 57 ratio larger than 2, as reported previously for primary COA (Reyes-Villegas et al., 2018; Sun et al., 2011; Cao et al., 2018; Sun et al., 2016). These characteristics distinguish CI-SV-OOA from other SV-OOAs reported in the literature, and thus, we consider that the classification of this factor simply as SV-OOA would provide only a partial description of its character.

Although, as noted by the reviewer, an evening peak would provide further evidence of the influence of cooking activities on CI-SVOOA, it is worth noting that this factor corresponds to atmospherically processed OA (O:C 0.61) and therefore, its diurnal behavior is not expected to resemble that of primary COA.

We have included additional text in the manuscript to provide further support for the denomination of this factor as CI-SV-OOA (lines 28-37, page 8).

Referee #1 specific comment 3: Page 10, line 19: In your CMB results you have said that the unclassified OC is likely SOA, would you expect more SOA at night (49%) compared to daytime (29%)? As you have already apportioned Biogenic and anthropogenic SOA in the model, I am guessing that this SOA is regional in nature, and so I

would not expect such a difference day/night.

Response to Referee #1 specific comment 3: We thank the reviewer for making this point, as it indicates that further clarification and explanation are needed. First, we have revised this description to include the unapportioned OC on an absolute scale to account for the 33% higher concentration of OC during daytime compared to nighttime. Second, we have added possible explanations for higher SOA at nighttime. The revised text reads (line 41, page 10 to line 4 page 11): "Notably, a substantial amount of OC was unapportioned, averaging 0.68 micro-gC m-3 (29%) in the daytime and 0.86 micro-gC m-3 (49%) in the nighttime... The higher unapportioned OC levels at night may be due to nighttime SOA formation (e.g., organonitrates formed by nitrate-radical initiated reactions) and/or to a shift in gas-particle partitioning to the particle phase with lower nighttime temperatures."

Works Cited

Al-Naiema, I. M., and Stone, E. A.: Evaluation of anthropogenic secondary organic aerosol tracers from aromatic hydrocarbons, Atmos. Chem. Phys., 17, 2053-2065, 10.5194/acp-17-2053-2017, 2017.

Cao, L.-M., Huang, X.-F., Li, Y.-Y., Hu, M., and He, L.-Y.: Volatility measurement of atmospheric submicron aerosols in an urban atmosphere in southern China, Atmos. Chem. Phys., 18, 1729-1743, 10.5194/acp-18-1729-2018, 2018.

Elser, M., Huang, R. J., Wolf, R., Slowik, J. G., Wang, Q., Canonaco, F., Li, G., Bozzetti, C., Daellenbach, K. R., Huang, Y., Zhang, R., Li, Z., Cao, J., Baltensperger, U., El-Haddad, I., and Prévôt, A. S. H.: New insights into PM2.5 chemical composition and sources in two major cities in China during extreme haze events using aerosol mass spectrometry, Atmos. Chem. Phys., 16, 3207-3225, 10.5194/acp-16-3207-2016, 2016.

Liu, T., Li, Z., Chan, M., and Chan, C. K.: Formation of secondary organic aerosols from gas-phase emissions of heated cooking oils, Atmos. Chem. Phys., 17, 10.5194/acp-

17-7333-2017, 2017.

Mohr, C., Huffman, J. A., Cubison, M. J., Aiken, A. C., Docherty, K. S., Kimmel, J. R., Ulbrich, I. M., Hannigan, M., and Jimenez, J. L.: Characterization of Primary Organic Aerosol Emissions from Meat Cooking, Trash Burning, and Motor Vehicles with High-Resolution Aerosol Mass Spectrometry and Comparison with Ambient and Chamber Observations, Environmental Science & Technology, 43, 2443-2449, 10.1021/es8011518, 2009.

Mohr, C., DeCarlo, P. F., Heringa, M. F., Chirico, R., Slowik, J. G., Richter, R., Reche, C., Alastuey, A., Querol, X., Seco, R., Peñuelas, J., Jiménez, J. L., Crippa, M., Zimmermann, R., Baltensperger, U., and Prévôt, A. S. H.: Identification and quantification of organic aerosol from cooking and other sources in Barcelona using aerosol mass spectrometer data, Atmos. Chem. Phys., 12, 1649-1665, 10.5194/acp-12-1649-2012, 2012.

Park, C., Schade, G. W., and Boedeker, I.: Characteristics of the flux of isoprene and its oxidation products in an urban area, J. Geophys. Res., 116, 10.1029/2011JD015856, 2011.

Reyes-Villegas, E., Bannan, T., Le Breton, M., Mehra, A., Priestley, M., Percival, C., Coe, H., and Allan, J. D.: Online Chemical Characterization of Food-Cooking Organic Aerosols: Implications for Source Apportionment, Environmental Science & Technology, 52, 5308-5318, 10.1021/acs.est.7b06278, 2018.

Srivastava, D., Favez, O., Bonnaire, N., Lucarelli, F., Haeffelin, M., Perraudin, E., Gros, V., Villenave, E., and Albinet, A.: Speciation of organic fractions does matter for aerosol source apportionment. Part 2: Intensive short-term campaign in the Paris area (France), Science Of The Total Environment, 634, 267-278, 10.1016/j.scitotenv.2018.03.296, 2018a.

Srivastava, D., Tomaz, S., Favez, O., Lanzafame, G. M., Golly, B., Besombes, J. L.,

Alleman, L. Y., Jaffrezo, J. L., Jacob, V., Perraudin, E., Villenave, E., and Albinet, A.: Speciation of organic fraction does matter for source apportionment. Part 1: A one-year campaign in Grenoble (France), Science Of The Total Environment, 624, 1598-1611, 10.1016/j.scitotenv.2017.12.135, 2018b.

Sun, Y., Du, W., Fu, P., Wang, Q., Li, J., Ge, X., Zhang, Q., Zhu, C., Ren, L., Xu, W., Zhao, J., Han, T., Worsnop, D. R., and Wang, Z.: Primary and secondary aerosols in Beijing in winter: sources, variations and processes, Atmos. Chem. Phys., 16, 10.5194/acp-16-8309-2016, 2016.

Sun, Y. L., Zhang, Q., Schwab, J. J., Demerjian, K. L., Chen, W. N., Bae, M. S., Hung, H. M., Hogrefe, O., Frank, B., Rattigan, O. V., and Lin, Y. C.: Characterization of the sources and processes of organic and inorganic aerosols in New York city with a high-resolution time-of-flight aerosol mass apectrometer, Atmos. Chem. Phys., 11, 1581-1602, 10.5194/acp-11-1581-2011, 2011.

Wallace, H. W., Sanchez, N. P., Flynn, J. H., Erickson, M. H., Lefer, B. L., and Griffin, R. J.: Source apportionment of particulate matter and trace gases near a major refinery near the Houston Ship Channel, Atmospheric Environment, 173, 16-29, https://doi.org/10.1016/j.atmosenv.2017.10.049, 2018.

Yan, B., Zheng, M., Hu, Y., Lee, S., Kim, H., and Russell, A.: Organic composition of carbonaceous aerosols in an aged prescribed fire plume, Atmos. Chem. Phys., 8, 6381-6394, 2008.
* * *
[Figure]

[Figure]

**Figure R1.** Time series of concentration of CI-SV-OOA and C$_3$H$_3$O$^+$ during the field campaign.

**Fig. 1.**

**Table R1**. Correlation between CI-SV-OOA and mass fragments previously reported as tracers of food cooking activities

| Mass fragment | Coefficient of correlation (R) | Reference(s) |
|:---:|:---:|:---:|
| $C_3H_3O^+$ | 0.89 | (Mohr et al., 2012) (Sun et al., 2016) (Wallace et al., 2018) |
| $C_2H_3O^+$ | 0.88 | (Mohr et al., 2009) (Liu et al., 2017) |
| $C_5H_8O^+$ | 0.73 | (Sun et al., 2016) (Sun et al., 2011) |
| $C_2H_4O_2^+$ | 0.70 | (Mohr et al., 2009) |
| $C_6H_6O^+$ | 0.75 | (Wallace et al., 2018) |
| $C_6H_{10}O^+$ | 0.51 | (Elser et al., 2016) (Cao et al., 2018) (Sun et al., 2016) (Sun et al., 2011) |

**Fig. 2.**

---

## Author Comment (AC3) · 1 Sep 2018

Referee #2 general comments: This manuscript reports a compositional analysis of PM2.5 by offline filter measurement on a day-night basis and NR-PM1 by an in-situ HR-TOF-AMS near the HSC in May 2015. Integrated source apportionment analysis for OC was conducted incorporating AMS-PMF, MM-PMF and CMB methods. The three source apportionment models are in agreement that ∼50% of OC was formed from primary fossil fuel combustion. SOA was further apportioned to anthropogenic and biogenic sources by combing results from MM-PMF and CMB. The comprehensive nature of this work in data and source apportionment analysis bring to the literature

a valuable case study report. This work nicely contributes to our improved understanding of SOA sources. I have the following specific points that need clarification/more discussion in their next revision.

Response to Referee #2 general comments: We thank the reviewer for their careful review of the manuscript and suggestions to improve it. We have responded point-by-point to the specific suggestions below.

Referee #2 major comment 1: The offline and online measurements overlapped between 13-27 May. Some comparison of the two measurements would be useful as data cross checking and verification. Such comparisons are currently lacking. For example, what's the percentage of PM1 contributed to total PM2.5 and how the two mass concentrations correlated with each other? This is important not only for data validation, but also for the latter comparison between AMS-PMF and MM-PMF and CMB.

Response to Referee #2 major comment 1: As suggested by the reviewer, we have added a comparison of AMS and filter-based measurements. We focus on three major species common to both sets of measurements. We do not compare PM mass, because PM mass was not measured on filters (instead by TEOM). The organic matter estimated from filters and organic aerosol measured by AMS is the most relevant to the comparison of source apportionment results. The following text has been added to section 3.2 on page 7 at lines 25-32:

"Filter-based PM2.5 measurements indicate the same major PM species as AMS NR-PM1 measurements and their ambient concentrations are compared here. The linear regression of the filter-based estimate of PM2.5 OM and NR-PM1 AMS OA had a slope of 0.61 and a low, but significant correlation (r = 0.48, p = 0.005), indicating that more OM was captured by the filter-based measurements than by the AMS. Sulfate measured by both techniques correlated strongly (r = 0.90, p < 0.001), with a slope of 0.89 indicating only a minor increase in filter-based sulfate relative to the AMS. Ammonium

correlated moderately (r = 0.72, p < 0.001) with a slope of 0.73. The consistently lower NR-PM1 concentrations measured by AMS relative to filters suggests the presence of OA, sulfate, and nitrate in the 1-2.5 micron size range and/or refractory matter that was not captured by the AMS."

Referee #2 major comment 2: There are inconsistencies among the source apportionment results by the different approaches. AMSPMF resolved a cooking influenced factor but no BB related OA, and the authors proposed the loss of the m/z 60 signatures during transport as possible reason. This reason is not supported by their data, as MM-PMF resolved both BB and cooking sources and BB even contributed more than cooking emission (11% vs 1%). It is odd that the authors didn't include cooking emission profiles in CMB. Previous studies have successfully apportioned PM2.5 to cooking emission (6%) in Fraser e al. [2003], as mentioned in the introduction part.

Response to Referee #2 major comment 2: We agree with the reviewer that the results from different source apportionment approaches could seem inconsistent. However, inherent differences between how the factors are resolved in each technique should be considered. In the case of AMS-PMF, 3 factors were selected as the main contributors to the observed PM1 concentrations. This selection was supported by evaluating variations in Q/Qexp and residual levels as additional factors were added to the model (SI section). Similar to the 3-factor solution, the solution containing 4 factors did not resolve a BB factor. As stated in the manuscript (line 25, page 13), the CI-SV-OOA factor exhibited low but statistically significant correlation with CMB BB, indicating potential inclusion of BB signatures in this factor. This is also evidenced by (i) a coefficient of correlation of 0.67 between the CI-SV-OOA and m/z 60 concentration time series and (ii) a CI-SV-OOA m/z 60 fraction (f60) of 0.003, the largest among the 3 factors retained in the AMS-PMF model. Although f60 in CI-SV-OOA falls at the lower edge of the region defined by Gilardoni et al (2016) to consider a factor influenced by BB, its proximity to this region indicates potential association between CI-SV-OOA and BB. These observations suggest that although BB signatures are likely present in the CI-

SV-OOA factor, AMS-PMF cannot effectively resolve these signatures, while MM-PMF is able to clearly separate the contributions of BB and food cooking. This reinforces the notion of complementarity between mass apportionment techniques and highlights the advantages associated to studying aerosol concentrations by employing simultaneous measurement and analyses techniques. Additional text including further reference to the potential inclusion of BB signatures in the CI-SV-OOA factor has been included in the manuscript (lines 29-34, page 13).

In regards to the CMB analysis, the reviewer is correct that a food cooking profile was not included and consequently this source was not apportioned by CMB. This choice was made in recognition that food cooking emissions vary greatly by type, style, and temperature of cooking. CMB-based source apportionment that relies upon a single, fixed source profile is ill equipped to capture this variability and the cooking signatures affecting ambient PM in HSC. Meanwhile, PMF is well suited to this task as it derives factor profiles from ambient measurements. Consequently we rely upon the MM-PMF result as a more accurate estimate of food cooking contributions to OC.

Referee #2 major comment 3: The AMS-PMF resolved a CI-SV-OOA factor that contributed to 31% of ambient OC (Table 5), but such a considerable contribution seems inconsistent with the minor cooking contribution estimated from MM-PMF (1%). This inconsistency calls into question whether naming the SV-OOA factor as "cooking-influenced" is appropriate, as it may imply this factor is largely influenced by cooking emissions.

Response to Referee #2 major comment 3: As stated in our answer to the previous comment, we agree with the reviewer that the contributions from cooking to OC derived from the different source apportionment approaches could seem inconsistent. However, it is worth noting that recognizing the influence of food cooking on the CI-SV-OOA factor is not equivalent to classifying this factor as cooking organic aerosol (COA). As included in our response to Referee #1, although the primary character of CI-SV-OOA would be mostly reflected by classifying this factor simply as atmospherically

processed aerosol with a semi-volatile character, this denomination would omit relevant mass spectral signatures of the CI-SV-OOA. To increase clarity, we have added some text in the manuscript emphasizing the fact that CI-SV-OOA is not to be confused with COA (lines 28-32, page 8). Additional reasoning behind the classification of this factor as CI-SV-OOA is included in our response to Referee #1 and has been added to the manuscript (lines 32-37, page 8).

Referee #2 major comment 4: The identification of different vehicle emission factors, i.e. diesel engines, gasoline engines and nontailpipe vehicle emissions, is not very convincing, considering alkanes, PAHs are not unique source tracers to vehicle emissions. It is unusual that norhopane and hopane are separated into two different factors. The two species are usually highly correlated with each other. What's the correlation between the two species in this study? Also, a high amount of EC was present in low-NOx anthropogenic SOA factor. Does this indicate the mixing of primary sources in this factor?

Response to Referee #2 major comment 4: We agree with the Referee that the diesel emissions, gasoline emissions and non-tailpipe emissions cannot be distinguished based on alkanes and PAH's, which are key chemical species in these factors. While these chemical species are indicative of a vehicle source, these are not unique tracers for diesel, gasoline, or non-tailpipe emissions. The identification of these three sources is thus based on EC:OC or PAH ratios, which are distinctive among these sources. For example, the EC:OC ratio of diesel engine emissions is greater than the EC:OC ratio of gasoline engine emissions and within the range of the values reported for these sources in the literature. Whereas, the absence of EC in the factor identified as non-tailpipe emissions is indicative of a non-combustion vehicle source. Furthermore, the ratio of benzo(a)pyrene to the sum of benzo(a)pyrene and chrysene in this factor is similar to the non-tailpipe emissions reported in the previous literature. In addition, the identification of the diesel and gasoline engines is supported by their correlations with the corresponding CMB source contributions. The separation of norhopane and

hopane into two separate factors can be explained by their weak correlation (r = 0.239, p = 0.110). This result suggests that two or more sources of hopane were present, each having different hopane ratios. We agree with the Referee that the presence of EC in the low-NOx anthropogenic factor is indicative of some mixing with primary fossil fuel combustion sources. To clarify this point, the following text has been added to lines 2-6 on page 13: "EC is also present in this factor, suggesting some mixing of this factor with combustion sources. Such mixing likely arises from VOC and precursors of oxidants co-emitted with EC from combustion contributing to SOA formation. However, the predominance of secondary organic markers over signatures of primary emissions suggests that this factor primarily represents SOA."

Referee #2 major comment 5: In the CMB model (section 2.5), were EC and levoglucosan included in the calculation? If not, which species was/were mainly responsible for determining the contributions from diesel engines and biomass burning? Also, it would be better if the statistical performance of the CMB results can be reported (e.g., $\chi 2$, calculated vs. modelled species concentrations, species source contributions).

Response to Referee #2 major comment 5: EC and levoglucosan were included in the CMB model calculation. While sitostanes, cholestane, and some PAH were included in the input data files, they were not used in the final model calculation, because some of these species were not available for the source profiles utilized. We have revised the methods section 2.5 to reflect the species included in the final model calculation (lines 2-5, page 6): "Species included in the CMB model included EC, levoglucosan, $17\alpha$(H)-$21\beta$(H)-hopane, $17\alpha$(H)-22,29,30-trisnorhopane, $17\beta$(H)-$21\alpha$(H)-30-norhopane, PAH (benzo(b)fluoranthene, indeno(1,2,3-cd)pyrene, and benzo(ghi)perylene), isoprene SOA tracers (2-methylglyceric acid and 2-methyltetrols), one $\alpha$-pinene SOA tracer (cis-pinonic acid), one naphthalene SOA tracer (phthalic acid), and one toluene SOA tracer (2,3-dihydroxy-4-oxopentanoic acid)."

As suggested by the reviewer, a summary of the model diagnostics have been added to the supporting information as Table S6: R2, $\chi$ 2, calculated-to-measured ratios for fitting species (EC, levoglucosan, 17$\alpha$(H)-21$\beta$(H)-hopane, 17$\alpha$(H)-22,29,30-trisnorhopane, 17$\beta$(H)-21$\alpha$(H)-30-norhopane, benzo(b)fluoranthene, indeno(1,2,3-cd)pyrene, and benzo(ghi)perylene). The following text has been added to section 3.4.1 (line 26 page 9): "CMB model diagnostics, including R2, $\chi$ 2, calculated-to-measured ratios for fitting species are summarized in Table S6."

The sample-by-sample CMB model results, including source contributions, the associated standard errors, measured and modeled OC, R2, and $\chi$ 2 values will be archived in an open access data base, along with the other data products linked to this manuscript.

Referee #2 major comment 6: The sample size of MM-PMF for PM2.5 OC is 46 in the study, but in the introduction section it is suggested the sample size should be 60–200 (P.2 line 39–40). It seems, because of this reason, the MMPMF performance is not very robust as shown in Table S2 (some key species like cis-pinonic acid and phthalic acid are not well modelled) and Table S5 (only 54% of bootstrap BB factor was mapped). The uncertain BB factor contribution in MM-PMF would also weaken the reliability of BB SOA estimation in this study. Please comment on how the small sample size affect the MM-PMF results in this work.

Response to Referee #2 major comment 6: We thank the reviewer for pointing this out and clarify the introduction point with a more up-to-date reference. The revised text reads (lines 2-4, page 3): "While PMF requires a relatively large sample size, MM-PMF has generated stable solutions with as little as 35 observational data points, which is expected to arise from the high specificity of primary and secondary source tracers Hu et al., (2010)."

We agree with the reviewer that there is uncertainty associated with the BB and BB SOA source estimates. We have added the following text to section 3.4.2 in introducing this factor (line 17-18, page 12): "Notably, only 54% of the bootstrap BB factor was mapped (Table S7; the lowest of any factor) indicating a greater relative uncertainty associated with this factor." We have also added the following text to section 3.5 (lines

3-6, page 14): "The BB SOA estimate is considered to be a best-estimate with the available data set, but contains uncertainties both from the MM-PMF and CMB estimates. Strategies to reduce the relative uncertainty associated with this source, include using a larger number of observations and more specific BB SOA tracers in MM-PMF."

Referee #2 major comment 7: Fig. 8 shows that the non-tailpipe vehicle emissions (marked largely by $17\alpha(H),21\beta(H)$-hopane only) made a notable contribution after 20 May (except 10 May daytime), is there any reason for it?

Response to Referee #2 major comment 7: The temporal variation of this factor is influenced by the detection of $17\alpha(H),21\beta(H)$-hopane. This species was detected only in 17 of the 46 PM2.5 samples analyzed, including 10 May in the daytime and 16 samples collected from 20-27 May.

Referee #2 major comment 8: Is there any reason to select $17\beta(H)$-$21\alpha(H)$-30-norhopane over $17\alpha(H),21\beta(H)$-30-norhopane as the vehicle emissions tracer in CMB and MM-PMF? The former species should be less abundant in ambient and source PM, and therefore is less commonly used in receptor models (e.g., Yu et al., 2011, Analytical and Bioanalytical Chemistry, 401, 3125-3139).

Response to Referee #2 major comment 8: We thank the reviewer for pointing this out. We have indeed used $17\alpha(H)$-$21\beta(H)$-30-norhopane in this study. We have corrected this typo throughout the manuscript, figures, tables, and supporting information.

Referee #2 major comment 9: When the source apportionment results from AMS-PMF, MM-PMF and CMB were integrated to obtain insights in SOA contributions (section 3.5), it seems the conclusion that anthropogenic SOA is the dominant contributor is largely drawn from the MM-PMF results, while the CMB and AMS-PMF results do not converge to the same conclusion (e.g., P.14 line 17: CMB is unable to provide a reliable estimation of total anthropogenic SOA, . . . ; and line 20–21: AMS-PMF is unable to distinguish between anthropogenic and biogenic origins of SOA, . . .). However, the MM-PMF results may have large uncertainties, especially considering the small sample

size and poor stability (e.g., only 67% of bootstrap high-NOx anthropogenic SOA factor can be mapped as shown in Table S5), making such a conclusion less convincing. Additional supportive evidence/argument for this point is recommended.

Response to Referee #2 major comment 9: The three models converge on the conclusion that anthropogenic SOA is a major source of ambient OC in HSC, with this finding led by the MM-PMF results and supported by the CMB and AMS-PMF results. In order to clarify this finding through additional evidence and argument, we have revised the last paragraph in section 3.5 (beginning on line 42, page 14):

"MM-PMF holds the advantage of identifying anthropogenic SOA for several reasons. First, these estimates are based on molecular tracers selective to this source (Al-Naiema and Stone, 2017). Second, these estimates were developed from ambient measurements within the HSC airshed and are considered to be the best representation of anthropogenic SOA in this location. Unlike CMB, MM-PMF requires neither a priori knowledge of tracer-to-OC ratios nor the assumption that these ratios are constant across chamber experiments and the study site. Third, by analyzing the co-variation of species over time, MM-PMF can capture anthropogenic SOA from other precursors that co-vary in time, even if they have a different precursor and are not defined by tracers in the source apportionment model (e.g., alkanes).

Nonetheless, CMB results support that anthropogenic SOA is an important source of OC in the HSC. Using available profiles for anthropogenic SOA in CMB, 3% of OC was attributed to monoaromatic-derived SOA (e.g., benzene, toluene) and 5% was attributed to naphthalene-derived SOA. Larger SOA contributions from PAHs compared to monoaromatic precursors agrees with controlled chamber photo-oxidation of diesel exhaust, in which PAH-derived SOA was estimated to account for up to 54% to the total SOA mass formed in the first 12 h of oxidation (Chan et al., 2009). The total PAH contribution to SOA in HSC is expected to be larger, since other 2-3 ring PAHs with SOA-forming potential (Shakya and Griffin, 2010) are co-emitted with naphthalene; however, molecular tracers for larger PAH oxidation products are not yet defined, so

the associated OC is unapportioned by the CMB model. Because the CMB-based estimate of anthropogenic SOA is limited to two classes of VOC precursors—aromatics and naphthalene derivatives—for which SOA profiles are available it is considered to be only a partial estimate of anthropogenic SOA. Anthropogenic SOA from other precursors contribute to the CMB unapportioned OC. The CMB-estimates of aromatic and naphthalene-derived SOA is valuable, however, because it provides specificity in the relative and absolute source contributions for these VOC classes.

The AMS-PMF LV-OOA factor that accounts for an average of 32% of PM1 OC is expected to be influenced by anthropogenic SOA based on the correlation observed with high-NOx anthropogenic SOA in MM-PMF (r=0.515, p=0.004). The MM-PMF and CMB methods that rely on specific tracers overcome the AMS limitation of being unable to distinguish between anthropogenic and biogenic origins of SOA in the absence of a strong signal that can identify a specific VOC precursor (Wallace et al., 2018; Xu et al., 2015). Relying on MM-PMF as the best estimate of anthropogenic SOA, we estimate that this source contributes an average of 28% of PM2.5 OC, making it a major aerosol source in HSC second only to primary fossil fuel emissions. Led by MM-PMF and supported by CMB and AMS-PMF, these reveal the important contributions of anthropogenic SOA to PM2.5 OC in the HSC area of Houston."

Finally, we have added the following sentence regarding the results of the bootstrapping analysis (line 42, page 14): "The former source contribution is considered to be stable with > 80% of bootstraps matched, while the latter has a larger relative uncertainty with 67% of bootstraps matched (Table S7)." We have addressed the reviewer's concerns about the number of observations used in the MM-PMF model in response to Referee #2 Comment 6.

Minor comment 1: Line 4-5 in p7: "organic carbon", "elemental carbon" and "Organic matter" should be removed as the abbreviations have been introduced before. Check throughout the context to avoid redundant words.

Response to minor comment 1: We prefer to maintain the definitions of these abbreviations at the start of the results and discussion section. We have checked the remaining text and can confirm that they are not repeated in the text after this occurrence.

Minor comment 2: Include the measured OC concentrations in Figure 8, which can help visualize directly how MMPMF predicts the measured concentrations.

Response to minor comment 2: As suggested by the reviewer, we have added the measured OC concentrations to Figure 8 and updated the figure caption accordingly.

Minor comment 3: Line 16 in p1: Change "fine" to "ambient".

Response to minor comment 3: We have made this change as suggested by the reviewer.

Minor comment 4: Line 25 in p1 and line 11 in p2: "VOC" should be "VOCs".

Response to minor comment 4: We define VOC as volatile organic compounds (note: in the plural form) on page 1 at line 25, such that an additional "s" is not needed in subsequent occurrences. We have checked the manuscript to ensure that "VOC" is used consistently.

Minor comment 5: Line 21 in p2: "for" should be "from".

Response to minor comment 5: We have made this change as suggested by the reviewer.

Minor comment 6: Line 13-16 in p6: the correlations are R or R2?

Response to minor comment 6: We have clarified at line 16 on page 6 that these are "correlation coefficients (r)..."

Minor comment 7: Figure S10, correct the typo mistake for the y-axis.

Response to minor comment 7: We have corrected this alignment error in the revised manuscript.

Minor comment 8: P.5 line 36: Bituminous coal source profile was said to be included in CMB, but it is not reported in the result section (3.4.1), please check.

Response to minor comment 8: The reviewer is correct that the bituminous coal source profile was tested in the CMB model; however, its source contributions were not statistically significant. We have added the following text to page 9 line 25: "The bituminous coal source contribution was not statistically significant."

Minor comment 9: P.8 line 25–26: The order of HOA, CI-SV-OOA and LV-OOA should be reversed to be consistent with the statement in the next sentence, as well as the abstract (line 20–22).

Response to minor comment 9: As suggested by the reviewer, we have revised the order of the AMS factors to be lowest to highest contribution. The revised text reads (line 1, page 8): "The time series of concentration of the AMS factors and their diurnal trends are presented in Fig. 5. The average mass concentrations observed for LV-OOA, CI-SV-OOA, and HOA during the field campaign were 0.37 ± 0.73 $\mu$g m-3, 0.48 ± 0.47 $\mu$g m-3, and 0.72 ±0.52 $\mu$g m-3, respectively, indicating a predominant contribution from secondary factors to the PM1 OA. The contribution to the OA mass concentration during the sampling period followed the sequence LV-OOA < CI-SV-OOA < HOA, with average abundances of approximately 22.33%, 29.44, and 48.23% respectively."

Minor comment 10: p.10 line 13: Full stop missing in "... comparing this study to Buzcu et al., (2006)."

Response to minor comment 10: We have corrected this error as suggested by the reviewer.

Minor comment 11: P.11 line 34–36: Yan et al. (2008) did not report isophthalic acid in aged BB plumes, please consider citing another reference.

Response to minor comment 11: Yan et al. (2008) reports in their table 2 (page 6390) that aromatic dicarboxylic acids are distinctively high in aged BB plumes compared to

non-aged BB, including 1,3-benzenedicarboxylic acid (a.k.a. isophthalic acid).

Minor comment 12: P.13 line 8 & 9: section 3.5.2 → section 3.4.2

Response to minor comment 12: We have corrected this text to read "section 3.4.2."

Minor comment 13: P.13 line 33–34 argued that isoprene-derived SOA contribution estimated by MM-PMF is more reliable than that by CMB, is there any further support/reference for this argument? Uncertainties associated with ambient measurements and temporal correlation between species in PMF should not be neglected.

Response to minor comment 13: We have clarified this point by adding the following text to section 3.5 (lines 27-30, page 14): "In contrast, CMB relies upon tracer-to-OC ratios observed in the laboratory (Kleindienst et al., 2007) where reactant concentrations greatly exceeded those observed in HSC (Figure S4). In particular, chamber concentrations were approximately 10-fold higher for NOx, 2-3 orders of magnitude higher for isoprene, and 3-4 orders of magnitude higher for toluene." Further, at line 25 on page 14, we have also added that the similar MM-PMF and CMB results "[indicate] good agreement between these two approaches."

Minor comment 14: Space missing: after Table S2 in P. 10 line 28; between "models reveals" in P.14 line 25; after NRPM1 in P.33 Table 2 title.

Response to minor comment 14: We have corrected these three errors as suggested by the reviewer.

Minor comment 15: P.14 line 32: . . ., from BB (5%), . . .

Response to minor comment 15: We have improved the readability of this sentence (line 39, page 15): "Together, these models were used to estimate primary sources of OC that included fossil sources (37-49%), BB (5%), and cooking (1%)."

Minor comment 16: P.15 line 5: . . . to organic aerosol by can be estimated by. . .

Response to minor comment 16: We have improved the readability of this sentence

(line 35, page 16): "For instance, BB SOA contributions to organic aerosol can be estimated by. . ."

Minor comment 17: Missing close parenthesis for graph (a) and (b) in Fig. S3.

Response to minor comment 17: The parentheses have been added as suggested by the reviewer.

Minor comment 18: Misaligned unit in the y-axis title of Fig. S10.

Response to minor comment 18: We have corrected this misalignment in the revised manuscript.

Works Cited

Al-Naiema, I. M., and Stone, E. A.: Evaluation of anthropogenic secondary organic aerosol tracers from aromatic hydrocarbons, Atmos. Chem. Phys., 17, 2053-2065, 10.5194/acp-17-2053-2017, 2017.

Chan, A. W. H., Kautzman, K. E., Chhabra, P. S., Surratt, J. D., Chan, M. N., Crounse, J. D., Kuerten, A., Wennberg, P. O., Flagan, R. C., and Seinfeld, J. H.: Secondary organic aerosol formation from photooxidation of naphthalene and alkylnaphthalenes: implications for oxidation of intermediate volatility organic compounds (IVOCs), Atmospheric Chemistry and Physics, 9, 3049-3060, 2009.

Gilardoni, S., Massoli, P., Paglione, M., Giulianelli, L., Carbone, C., Rinaldi, M., Decesari, S., Sandrini, S., Costabile, F., Gobbi, G.P., Pietrogrande, M.C., Visentin, M., Scotto, F., Fuzzi, S., Facchini, M.C., Direct observation of aqueous secondary organic aerosol from biomass-burning emissions. Proceedings of the National Academy of Sciences 113, 10013-10018. 10.1073/pnas.1602212113, 2016.

Hu, D., Q. Bian, A. K. H. Lau and J. Z. Yu, Source apportioning of primary and secondary organic carbon in summer PM2.5 in Hong Kong using positive matrix factorization of secondary and primary organic tracer data. Journal of Geophysical Research:

[Figure]

Atmospheres 115 (D16), D16204, doi.10.1029/2009JD012498, 2010.

Kleindienst, T. E., Jaoui, M., Lewandowski, M., Offenberg, J. H., Lewis, C. W., Bhave, P. V., and Edney, E. O.: Estimates of the contributions of biogenic and anthropogenic hydrocarbons to secondary organic aerosol at a southeastern US location, Atmos. Environ., 41, 8288-8300, 10.1016/j.atmosenv.2007.06.045, 2007.

Shakya, K. M., and Griffin, R. J.: Secondary Organic Aerosol from Photooxidation of Polycyclic Aromatic Hydrocarbons, Environmental Science & Technology, 44, 8134-8139, 10.1021/es1019417, 2010.

Wallace, H. W., Sanchez, N. P., Flynn, J. H., Erickson, M. H., Lefer, B. L., and Griffin, R. J.: Source apportionment of particulate matter and trace gases near a major refinery near the Houston Ship Channel, Atmos. Environ., 173, 16-29, doi.org/10.1016/j.atmosenv.2017.10.049, 2018.

Xu, L., Guo, H. Y., Boyd, C. M., Klein, M., Bougiatioti, A., Cerully, K. M., Hite, J. R., Isaacman-VanWertz, G., Kreisberg, N. M., Knote, C., Olson, K., Koss, A., Goldstein, A. H., Hering, S. V., de Gouw, J., Baumann, K., Lee, S. H., Nenes, A., Weber, R. J., and Ng, N. L.: Effects of anthropogenic emissions on aerosol formation from isoprene and monoterpenes in the southeastern United States, Proc. Natl. Acad. Sci. U.S.A., 112, 37-42, 10.1073/pnas.1417609112, 2015.

Yan, B., Zheng, M., Hu, Y., Lee, S., Kim, H., and Russell, A.: Organic composition of carbonaceous aerosols in an aged prescribed fire plume, Atmos. Chem. Phys., 8, 6381-6394, 2008.
* * *
**Table S6**: CMB model performance metrics. The $R^2$ values indicate the fit of the profile to the ambient data, with values greater than 0.8 indicating a good model fit. The $\chi^2$ values are the weighted sum of squares of the differences between the calculated and measured fitting species concentrations, with a value of 0 indicating a perfect model fit, <1 indicating a very good fit, 1-2 indicating an acceptable fit, and >4 indicating that one or more species concentrations are not well explained by the model. In one sample (21 May, nighttime) this value was greater than 4, with 2.85 the next-highest value. The calculated-to-measured concentration ratios of the fitting species indicate the extent to which individual tracers were fit by the model. SOA tracers, which behaved ideally coming from only one source, had calculated-to-measured concentrations of 1.

| Performance Metric | Range | | | Mean | Median |
|---|---|---|---|---|---|
| $R^2$ | 0.833 | - | 0.999 | 0.979 | 0.987 |
| $\chi^2$ | 0.06 | - | 4.35 | 0.93 | 0.71 |
| Calculated-to-measured concentration ratios | | | | | |
| elemental carbon | 0.99 | - | 1.01 | 1.00 | 1.00 |
| levoglucosan | 0.61 | - | 1.32 | 0.98 | 0.99 |
| 17α(H)-21β(H)-hopane | 0.00 | - | 0.42 | 0.21 | 0.20 |
| 17α(H)-21β (H)-30-norhopane | 0.75 | - | 1.07 | 0.99 | 1.00 |
| 17α(H)-22,29,30-trisnorhopane | 0.67 | - | 1.33 | 0.99 | 1.00 |
| benzo(b)fluoranthene | 0.67 | - | 2.00 | 1.05 | 1.00 |
| benzo(ghi)perylene | 0.00 | - | 1.78 | 0.92 | 1.00 |
| indeno(1,2,3-cd)pyrene | 0.00 | - | 2.00 | 1.06 | 1.00 |

**Fig. 1.**